# Flow Equivariant World Models:
# Structured Memory for Dynamic Environments

Hansen Jin Lillemark [* 1 2]   Benhao Huang [* 3]   Fangneng Zhan [4]   Yilun Du [1]   T. Anderson Keller [1]

## Abstract

Embodied systems experience the world as 'a symphony of flows': a combination of many continuous streams of sensory input coupled to self-motion, interwoven with the dynamics of external objects. These sensory streams and the underlying dynamics of the world obey smooth, time-parameterized symmetries which existing world models ignore. Without a memory that respects this structure, partial observability presents a major obstacle to existing methods: each observation reveals only a fraction of the world, while unobserved regions continue to evolve. In this work, we introduce Flow Equivariant World Modeling, a framework that leverages time-parameterized symmetries within a latent memory for stable and accurate dynamics prediction over long horizons. The latent memory shifts and transforms equivariantly with self-motion and inferred external object motion, keeping information about out-of-view regions aligned as time progresses. We demonstrate the advantage of this framework over state-of-the-art diffusion, memory-augmented, and recurrent world model architectures on 2D and 3D partially observed video world modeling benchmarks. More broadly, our results suggest that predictive representations become more powerful when they are organized in line with the temporal and dynamical structure of the world they model. Project page: flowequivariantworldmodels.github.io

## 1. Introduction

As embodied agents in a dynamic world, our survival critically depends on our ability to accurately model our surrounding environment, our own self-motion through it,

*Equal contribution [1]Kempner Institute, Harvard University [2]CSE, UC San Diego [3]ML, Carnegie Mellon University [4]SEAS, Harvard University. Correspondence to: <hlillemark@ucsd.edu>, <benhaoh@andrew.cmu.edu>, <ydu@seas.harvard.edu>, <t.anderson.keller@gmail.com>.

*Proceedings of the 43rd International Conference on Machine Learning*, Seoul, South Korea. PMLR 306, 2026. Copyright 2026 by the author(s).

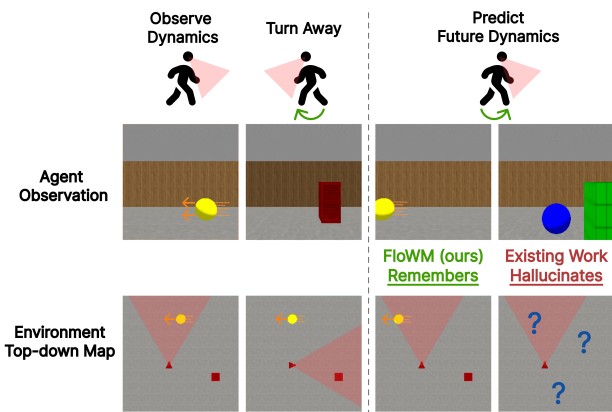

*Figure 1.* **Flow Equivariant World Models (FloWM) maintain structured dynamic memory in partially observed environments.** When an agent observes dynamics, turns away, then turns back to the original viewpoint, flow equivariance asserts dynamics continue even when unobserved; existing work loses track of objects and eventually hallucinates new objects in their place.

and the dynamics of moving bodies within it. An example from nature is pack hunting: to coordinate an attack, an agent must accurately estimate the location and velocity of a target while simultaneously predicting the motion of other pack animals. However, these world states are not simply provided to the agent in the form of an omniscient global view; instead, the agent only has a restricted first-person field of view that simultaneously shifts and rotates with the agent's own self-motion. The result is a highly entangled stream of stimulus flows that yield only a fraction of the full environment's information at any point in time. Despite this, biological agents appear to navigate such partially observed dynamic environments effortlessly, as if they have a memory of the environment that moves in unison with the global world state.

In this work, we formalize how such a geometrically structured dynamic memory may be built in service of the task of *partially observed dynamical world modeling* (visualized in Fig. 1). Specifically, we find that both internal and external motion can be understood as mathematical 'flows', enabling each source of variation to be handled exactly as time-parameterized symmetries through the framework of 'flow equivariance' (Keller, 2026). Flow Equivariant World Models can be constructed to handle self-generated mo-

tion in a precisely structured manner, while simultaneously capturing the motion of external objects, even if they are moving outside the observed field of view. We show that this yields substantially improved video world modeling performance and generalization to significantly longer sequences than those seen during training, highlighting the benefits of precise spatial and dynamical structure in world models.

## 2. Background

To build world models with structured representations of dynamic environments, we rely on recent work in both world modeling and equivariance, briefly reviewed here.

**World Modeling.** At a high level, a world model is a system affording the ability to predict not only the future state of an environment given initial conditions, but also how that state may evolve differently when acted upon by an agent (Ha & Schmidhuber, 2018). Recent work on world models has focused on representing and predicting the future world state as video, primarily using large-scale latent diffusion transformer models (Peebles & Xie, 2023; Wan et al., 2025). While these models achieve impressive perceptual quality and scale well with growing data and compute, their current form inherently lacks the ability to predict long-horizon dynamics, especially in partially observable environments (Figure 2).

**Partial Observability.** Defined as a setting when the agent's observation does not contain the full information of the world's state, partial observability is particularly relevant to 3D world modeling where agents inherently have a limited field of view. This limitation necessitates a form of memory in order to represent and integrate partial information through time. Recent autoregressive methods such as History-Guided Diffusion Forcing address this memory requirement by extending the transformer self-attention window over many past observation frames to retain self consistency (Song et al., 2025). Yet as the number of observation frames grows, information must be inevitably discarded through sliding-window attention or some other approximation. This problem is exacerbated by the cost of spatiotemporal attention over a highly redundant signal such as video. Once relevant past information has left the context window, it has been lost; turning around will reveal an entirely new hallucinated scene (Figure 1).

**Memory.** Recent work has explored augmenting video diffusion models with different forms of latent memory that persist across time; however, the focus has primarily been on consistency in static 3D scenes, without a unified framework for modeling partially observed dynamics (Po et al., 2025; Savov et al., 2026; Xiao et al., 2025; Zhou et al., 2026; Wu et al., 2026). In this work, we argue that a natural way to build world models is with a recurrent flow equivariant memory at the core, evolving and shifting to

represent both the dynamics of the world and the actions of the agent seamlessly. Such a memory enables prediction of future states in a structured manner while maintaining important information over unbounded timespans.

**Equivariance.** A neural network $\phi$ is said to be equivariant if its output, $\phi(f)$, changes in a structured, predictable manner when the input $f$ is transformed by an element $g$ of the group $G$, i.e. $\phi(g \cdot f) = g \cdot \phi(f) \ \forall g \in G$. One way to construct equivariant neural networks is through structured weight sharing (Cohen & Welling, 2016; Ravanbakhsh et al., 2017). This structure reduces the number of parameters that need to be learned in an artificial neural network while simultaneously improving performance by incorporating known symmetries from the data distribution. For example, in the setting of molecular dynamics simulation, introducing equivariance with respect to 3-dimensional translations, rotations, and reflections (a known symmetry of the laws of physics) increases data efficiency by up to three orders of magnitude (Batzner et al., 2022). More broadly, when the data-generating process respects a group symmetry, equivariance can yield a provably strict improvement in generalization (Elesedy & Zaidi, 2021). Our work aims to identify and incorporate time-parameterized symmetries to bring similar generalization benefits to world modeling.

## 3. Flow Equivariant World Models

In this section, we begin with a review of flow equivariance, and describe how it must be extended to account for partial observability and self-motion. In doing so, we will see how it naturally motivates a structured persistent memory, resulting in our general framework for partially observed dynamic world modeling.

### 3.1. Generalized Flow Equivariance

Flow equivariance extends existing 'static' group equivariance to time-parameterized sequence transformations ('flows'), such as 2D visual motion (Keller, 2026). These flows are generated by vector fields $\nu$, and written as $\psi_t(\nu) \in G$ (often dropping the dependence on $\nu$ to avoid clutter). The flow $\psi_t$ maps from some initial group element $g_0$ to a new element $g_t$ (i.e. $\psi_t(\nu) \cdot g_0 = g_t$) by following the vector field for $t$ timesteps. Formally, a flow $\psi_t(\nu) : \mathbb{R} \times \mathfrak{g} \to G$ is a subgroup of a Lie group $G$, generated by a corresponding Lie algebra element $\nu \in \mathfrak{g}$, and parameterized by a value $t \in \mathbb{R}$ often interpreted as time.

Consider a sequence of input signals $f = \{f_t\}_{t=0}^T$, where each timestep maps from an input space $X$ to $K$ channels: $f_t : X \to \mathbb{R}^K$. A sequence-to-sequence model $\Phi$ defines a map $f \mapsto h$ for outputs $h = \{h_t\}_{t=0}^T$, $h_t : X' \to \mathbb{R}^{K'}$, such as mapping from an input video to a sequence of hidden states. $\Phi$ is said to be *flow equivariant* if, when the input sequence undergoes a flow, i.e. $\{f_t\}_{t=0}^T \to \{\psi_t \cdot f_t\}_{t=0}^T$, the output sequence also transforms according to the action of

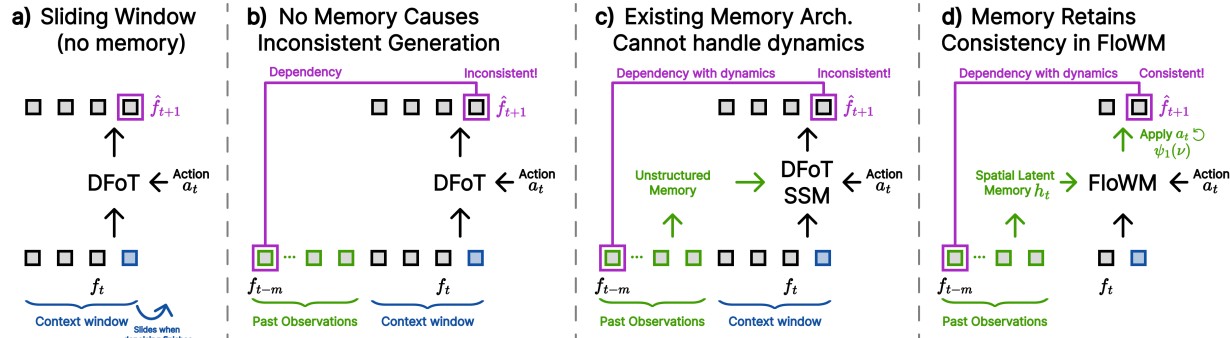

*Figure 2.* **Existing world model memory is inherently limited in partially observed dynamic environments.** a) Standard autoregressive video diffusion evicts frames beyond the sliding window. b) Information dependencies between past observations and generated frames cause inconsistency without memory. c) Existing memory solutions are view-dependent (unstructured), and thus cannot predict dynamic scenes consistently. d) FloWM remembers past observations in a structured memory that is continually updated via internal dynamics.

that flow $\{h_t\}_{t=0}^T \to \{\psi_t \cdot h_t\}_{t=0}^T$. Explicitly, as a relation:

$$\Phi\left(\{\psi_t \cdot f_t\}_{t=0}^T\right)_T = \psi_T \cdot \Phi\left(\{f_t\}_{t=0}^T\right)_T \ \forall T, \quad (1)$$

where the action of the flow on signal $f_t$ over the input space $X = G$ is defined via the pullback: $\psi_t \cdot f_t(g) := f_t(\psi_t^{-1} \cdot g)$. The group $G$ can be thought of here as the spatial coordinate of an image signal. Informally, flow equivariance asserts: if the input observation moves over time, the output should also move over time.

In the case of partial observability, flows are more naturally stated to operate on unobserved world state sequences $\{w_t\}_{t=0}^T \to \{\psi_t \cdot w_t\}_{t=0}^T$. The world state $w_t$ can be understood to encompass the entirety of the environment: what is directly in front of the agent, what is behind the agent, and what is in the next town over. The agent's restricted observations are then produced through an observation function $\mathcal{O}(w_t) = f_t$. In such a setting, world flows may not appear as exact flows in the observation sequences themselves (i.e. $\mathcal{O}(\psi_t \cdot w_t) \neq \psi_t \cdot f_t$). For example, a car approaching from behind you may have no immediate impact on what you see; however, as we will demonstrate, equivariance to such world flows is no less critical to accurately predicting future observations in a partially observed world – that car may come speeding past you in a few seconds.

Formally, we write global flow equivariance as:

$$\Phi\left(\{\mathcal{O}\left(\psi_t \cdot w_t\right)\}_{t=0}^T\right)_T = \psi_T \cdot \Phi\left(\{\mathcal{O}\left(w_t\right)\}_{t=0}^T\right)_T \ \forall T.$$

In practice, this means that we must now define the output of $\Phi$ to be equivariant with respect to the full world state, implying that the latent representation is structured to encode a faithful 'memory map' of the dynamic environment instead of just the current observation. Intuitively, since the hidden state must transform equally whether an object is in view or not, it must maintain a structured memory that mirrors the structure of the world.

To achieve flow equivariance generally, it is sufficient to perform computation in the co-moving reference frame of the input (Keller, 2026). Intuitively, if the contents of

the input signal move according to a flow $\psi_t(\nu)$, then the recurrent memory must be transported by the same flow before incorporating the next observation. This ensures that features stored in the hidden state remain spatially aligned with the moving input, rather than being left behind in a fixed reference frame. For a simple Recurrent Neural Network (RNN), this takes the form:

$$h_{t+\Delta t} = \sigma\left(\psi_{\Delta t}(\nu) \cdot h_t + f_t\right). \quad (2)$$

To achieve equivariance with respect to a set of multiple flows ($\nu \in V$), Flow Equivariant RNNs possess multiple hidden state 'velocity channels', each flowing according to their own vector fields $\nu$. The input space of the hidden state $h_t$ is thus $X' = V \times G$. Fig. 3 illustrates the different 'velocity channels' as individual stacked maps $h_t(\nu): G \to \mathbb{R}^{K'}$, where $G$ can be thought of as the spatial coordinate of each recurrent 'velocity channel'. Because the elements of the Lie algebra combine in a structured manner, it is then possible to show that when the input sequence is acted on by a flow $\psi(\hat{\nu}) = \{\psi_t(\hat{\nu})\}_{t=0}^T$, the hidden state outputs also flow, and the 'velocity channels' permute according to the difference between their velocity and the input velocity, $(\nu - \hat{\nu})$:

$$h_t[\psi(\hat{\nu}) \cdot f](\nu) = \psi_{t-1}(\hat{\nu}) \cdot h_t[f](\nu - \hat{\nu}) \ \forall t, \quad (3)$$

where $h_t[f]$ denotes passing $f$ to the RNN as an input.

**Generalized Flow Equivariant Recurrence Relation.** To support more complex tasks, such as 3D partially observed world modeling, we introduce an abstract version of the flow equivariant recurrence relation which supports arbitrary encoders and update operations. Specifically, we define our abstract observation encoder as $\mathrm{E}_\theta[f_t; h_t]$, a function of the current observation $f_t$ and the prior hidden state $h_t$; and we define our abstract recurrent update operation as $h_{t+1} = \mathrm{U}_\theta[h_t; o_t]$, a function of the encoded observation ($o_t = \mathrm{E}_\theta[f_t; h_t]$) and the past hidden state. The internal velocity channels flow for one timestep via the action of $\psi_1$. Putting them together, the new *generalized flow equivariant*

*recurrence relation* can then be written as:

$$h_{t+1}(\nu) = \psi_1(\nu) \cdot U_\theta \left[ h_t(\nu); \ E_\theta \left[ f_t; h_t \right] (\nu) \right]. \quad (4)$$

To prove that this is indeed still flow equivariant, both the encoder and update operations are required to be equivariant with respect to transformations on their inputs:

$$E_\theta \left[ g \cdot f_t; \ g \cdot h_t \right] = g \cdot E_\theta \left[ f_t; h_t \right] \quad (5)$$

$$U_\theta \left[ g \cdot h_t; \ g \cdot o_t \right] = g \cdot U_\theta \left[ h_t; \ o_t \right] \quad (6)$$

Secondly, we also require that the Encoder performs a 'trivial lift' of the input to all velocity channels, such that: $E_\theta \left[ f_t; h_t \right] (\nu) = E_\theta \left[ f_t; h_t \right] (\hat{\nu}) \ \forall \nu, \hat{\nu} \in \mathfrak{g}$. In Appendix Section A, we prove formally that this new framework indeed retains the flow equivariance properties of the original Flow Equivariant RNN for fully observed environments.

**Self-Motion Equivariance.** Finally, to complete our framework, we note that motion is relative (i.e. self-motion of an agent is equivalent to global motion of the input). Therefore, flow equivariance may be naturally extended to encompass not just motion of external objects, but also the self-generated motion of agents. Self-motion is usually known by the agent corresponding to the action taken between the agent's intervening observations. This additional information allows us to build a world model whose memory exists in the co-moving reference frame of the agent, thereby achieving self-motion equivariance, without adding any further 'velocity channels'; together, we call this model *FloWM*. Crucially, as a byproduct of this self-motion equivariance, the memory of the model becomes spatially structured in a manner homomorphic to the structure of the world. Intuitively, as the agent explores the world, it leverages its own intervening actions to integrate new observations into its memory in the correct relative positions, building a structured 'map' of its environment. This procedure is reminiscent of recent theories for how prefrontal cortex stores structured memories in the brains of primates (Whittington et al., 2025).

In detail, given the action variable $a_t$, denoting the action of the agent between times $t$ and $t+1$, we transform the hidden state of the network to flow according to the latent group representation of the action, denoted $T_{a_t}$, resulting in the following *Self-Motion Flow Equivariant* Recurrence Relation:

Next Latent Memory State    Internal Flow Transform    Latent Memory Defined Over $\nu$

$$\overbrace{h_{t+1}(\nu)} = \underbrace{T_{a_t}^{-1}} \overbrace{\psi_1(\nu)} \cdot \underbrace{U_\theta} \left[ \overbrace{h_t(\nu)}; \ \underbrace{E_\theta[f_t, h_t](\nu)} \right]. \quad (7)$$

Self Action Transform    Memory Update    Encoder Output

In the case when the action space is the 2D translation group (such as in our MNIST World experiments in the following section), the representation $T_{a_t}$ takes the form

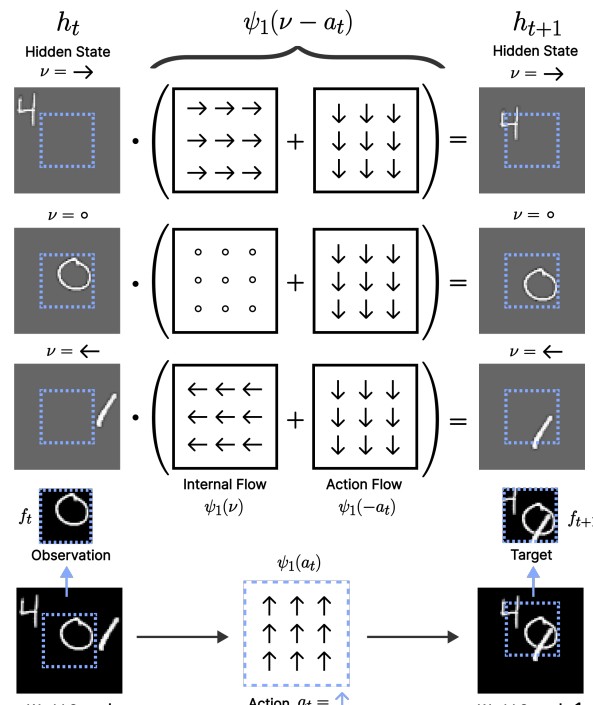

*Figure 3.* **Visualization of the Simple Recurrent FloWM on MNIST World.** The world state (bottom) is windowed to produce an observation $f_t$. The RNN processes this observation and embeds it in its hidden state composed of the 'velocity channels' (stacked vertically, of which there are three for visualization purposes). Self-motion and external motion flow equivariance are obtained by acting on the hidden state with the combination of the action flow $\psi_1(a_t)$ and each channel's corresponding flow $\psi_1(\nu)$.

of another 'action flow' ($\psi_1(-a_t)$) describing the visual flow induced by the action in the agent's reference frame. By the properties of flows, this then combines with the 'internal flows' ($\psi_1(\nu)$) of the 'velocity channels', yielding a simple combined flow, $\psi_1(\nu - a_t)$, in the recurrence (Figure 3). When the action space is more sophisticated, such as involving rotations in 3D environments, the representation acts directly on the spatial dimensions and velocity channels of the hidden state itself (e.g. rotating velocity vectors). In the following paragraphs we describe precisely how these abstract elements are instantiated for each of the datasets we explore in this study.

### 3.2. Instantiations for 2D / 3D Partially Observed Dynamic World Modeling

**Simple Recurrent FloWM.** For the first set of experiments, to validate our framework in a 2D environment, we construct a recurrent model with self-motion and flow equivariance following the framework introduced above. Explicitly:

$$h_{t+1}(\nu) = \psi_1(\nu - a_t) \cdot \sigma \left( \mathcal{W} \star h_t(\nu) + \mathrm{pad}(\mathcal{U} \star f_t) \right), \quad (8)$$

where $\mathcal{W} \star h_t$, and $\mathcal{U} \star f_t$ denote convolutions over the hidden state and spatial input observation. To model partial

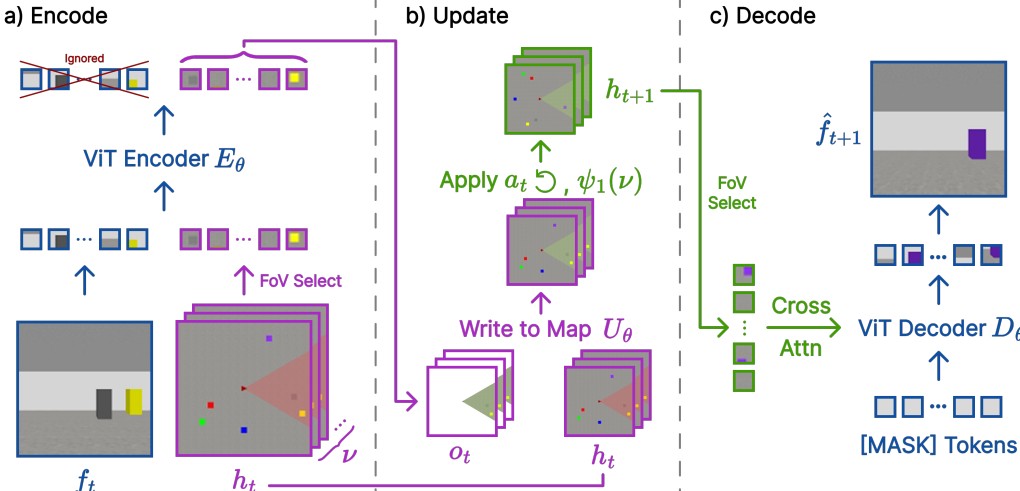

*Figure 4.* **Transformer-Based FloWM. a)** Image observation $f_t$ at time $t$ and memory tokens $h_t$ within the current field of view (FoV) are passed through a ViT encoder $E_\theta$. The latent map $h_t$ is fully learned, visualized as a map here for clarity. **b)** The update writes to the memory tokens at the correct FoV locations, then transforms the latent map according to known action $a_t$ and internal flow $\psi_1(\nu)$, producing $h_{t+1}$. **c)** We decode using cross attention over the new FoV tokens of $h_{t+1}$ with a ViT decoder $D_\theta$ to predict next image $\hat{f}_{t+1}$.

observability, we write to (denoted 'pad($\cdot$)'), and read from, a fixed window_size < world_size portion of the hidden state (blue dashed square in Fig. 3), letting the rest of the hidden state flow around the agent's field of view according to $\psi_1(\nu - a_t)$. In particular, the hidden state is windowed at each timestep, pixel-wise max-pooled over 'velocity channels' and passed through a decoder $g_\theta$ to predict the next observation, explicitly: $\hat{f}_{t+1} = g_\theta(\max_\nu(\text{window}(h_{t+1})))$. We see that this is an instantiation of our general framework with $E_\theta[f_t; h_t](\nu) = \mathcal{U} \star f_t$, and $U_\theta[h_t; o_t] = \sigma(\mathcal{W} \star h_t + \text{pad}(o_t))$ where all operations are equivariant to translation, and thus satisfy the conditions of Equations 5 & 6.

**Transformer-Based FloWM.** To extend our framework to more complex datasets, we construct a second FloWM instantiation with a Vision Transformer (ViT) (Dosovitskiy et al., 2021) encoder and decoder, depicted in Fig. 4. In this setting, the recurrent state $h_t$ is a set of spatially organized token embeddings that act as a structured latent map of the 3D world; the key difference is the additional expressivity endowed by the ViT per-step encoder and decoder. In the spirit of Ha & Schmidhuber (2018), we assume this map to be a 'top-down' 2D abstract version of the true 3D environment. We denote this set of tokens $h_t := \{h_t^{(x,y)} | (x, y) \in [0, W] \times [0, H]\}$ where $(x, y)$ are the spatial coordinates of the token. Importantly, this map is group-structured with respect to the agent's action group (2D translation and 90-degree rotation), and the group of external object motion (2D translation), giving us a known form of the representation of $T_{a_t}$ in the latent map (see Cohen & Welling (2016)).

The goal of the encoder $E_\theta[f_t; h_t]$, instantiated as a ViT, is then to take the tokens of the map corresponding to the current field of view, and update them using the image

patch tokens ($\text{patchify}(f_t)$). Explicitly: $o_t = E_\theta[f_t; h_t] = \text{ViT}[\text{concat}[\text{patchify}(f_t); \text{FoV}(h_t)]]$, where $\text{FoV}(h_t)$ returns a fixed subset of $h_t$ corresponding to the 2D triangular wedge field of view of the agent, depicted in Fig. 4 (red triangle). The update operation $U_\theta[h_t; o_t]$ then simply adds these updated view tokens to the latent map in the correct positions (see §G.4). The simple addition is indeed equivariant with respect to linear transformations of its inputs, satisfying Equation 6.

In order to satisfy Equation 5 formally, $E_\theta$ must be equivariant with respect to motion of the agent and external objects and perform a 'trivial-lift' to the velocity channels. However, since the encoder must map from the 3D first person point of view of the agent, to an abstract top-down map, it is highly non-trivial to make this transformation exactly equivariant by design without relying on explicit and costly depth unprojection. Therefore, instead, we simply treat the output of the encoder as if it were performing this equivariant lift, and anticipate that the transformation $T_{a_t}^{-1}\psi_1(\nu)$ between timesteps will encourage the encoder to learn to become equivariant, as has been demonstrated in prior work (Keller & Welling, 2021; Keurti et al., 2023). As demonstrated empirically in the following section, this holds with high fidelity in practice, and in fact is perhaps one of the greatest strengths of FloWM. Rather than maintaining costly 3D hidden state volumes and approximate 2D reductions required for exact encoder equivariance, this learned equivariance approach instead allows the powerful encoder to learn the optimal abstraction which obeys equivariance. In Section 5 we review related models that have similarly structured representations with respect to self-motion, but instead rely on exact encoder equivariance, thereby limiting their scalability. (See §G for full details).

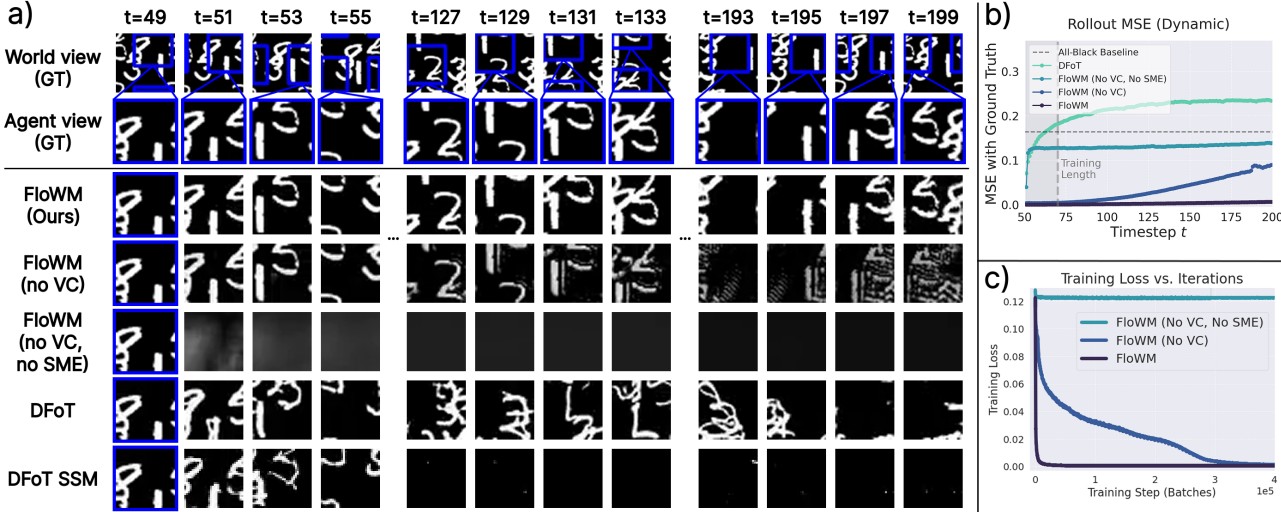

*Figure 5.* **FloWM generalizes further and trains faster. a)** Timesteps 0 to 49 are given as observations. Models are trained to predict up to $t = 69$. We see that FloWM rollouts do not diverge even at timestep 199, while baselines slowly degrade in image quality or lose track of the digits. **b)** MSE vs. rollout timestep shows length generalization. **c)** Learning efficiency of the FloWM vs. ablations.

## 4. Experiments

To address the scarcity of research on *partially observed world modeling*, we introduce a new suite of benchmarks and use them to compare our FloWM framework against state-of-the-art video diffusion and recurrent world models. Our results demonstrate that the structured dynamic memory afforded by flow equivariance is critical for modeling partially observed dynamic environments. Code is available on our project page.

### 4.1. Diffusion-based and Recurrent Baselines

**Diffusion Forcing Transformer.** Due to its claims of long term consistency, we compare with History-guided Diffusion Forcing (denoted DFoT) as our primary baseline, using latent diffusion with a CogVideoX-style transformer backbone (Song et al., 2025; Chen et al., 2024; Yang et al., 2025). Following standard practice, we first train a spatial downsampling VAE for each dataset to encode video frames into a latent representation that is then input to the diffusion model. The DFoT's training objective makes no distinction between observation and prediction frames, unlike FloWM, and trains on fixed length sequences in the self-attention window. To condition on actions, we follow CogVideoX-style action conditioning, including the embedded action sequence for all frames in the self-attention window. During inference, DFoT maintains a sliding window composed of context and prediction frames at different noise levels; after denoising is complete on one chunk, the sliding window advances (visualized in Figure 2(b)). Full DFoT details can be found in §K.2.

**Diffusion Forcing State Space Model.** As a representative baseline for memory-augmented video diffusion models, we compare against a recent approach denoted DFoT-

SSM (Po et al., 2025) that combines a short horizon Diffusion forcing transformer (for local consistency) with a blockwise scan State Space Model module (for long horizon memory) (Dao & Gu, 2024). This baseline is visualized in Figure 2(c). This model employs the same VAE encoder, a similar inference procedure, and the same action conditioning as DFoT (see §K.4 for more details).

**Recurrent State Space Model.** For the 3D Block World dataset, we additionally implement a Recurrent State Space Model (RSSM) based on Dreamer V3's world model, as a representative non-diffusion baseline (Hafner et al., 2023). The model's recurrent state is used to predict future observations given action conditions, but has no structured memory; full details are in §J.

### 4.2. 2D MNIST World Benchmark

**Dataset.** To test our architecture on partially observable dynamic world modeling, we propose a simple MNIST World dataset. The world is a 2D black canvas with multiple MNIST digits moving at random constant velocities. The agent is provided a view of the world, smaller than the world size, yielding partial observability. At each discrete timestep, the world evolves according to the velocity of each object, and the agent takes a random action (relative $(x, y)$ offset) to move its viewpoint. The edges roll, meaning a digit that moves off the screen to the left will reappear on the right. Given 50 observation frames, the training task is to predict the dynamics for 20 future frames, integrating the dynamics of the observed digits and the given self-motion actions. To test length generalization, we additionally run inference up to 150 prediction frames. We include ablations on data subsets with different combinations of partial observability, presence of object dynamics, and self-motion in §F.

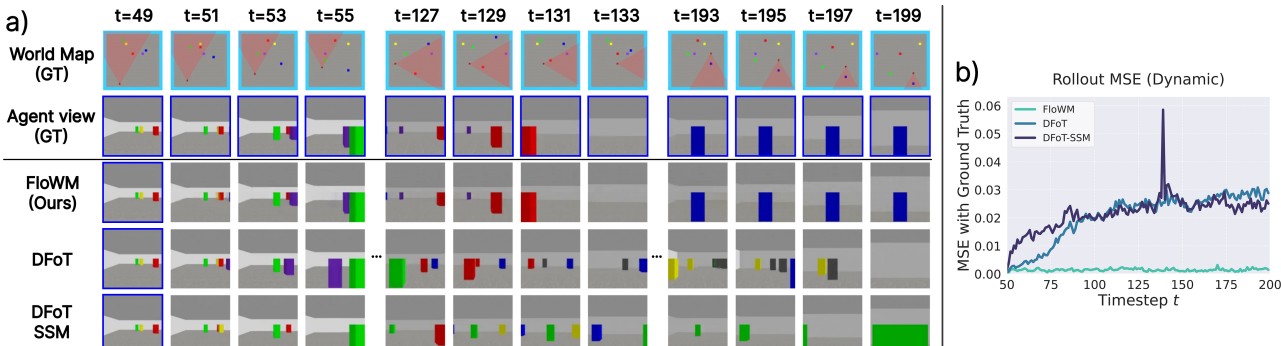

*Figure 6.* **Flow Equivariant World Models accurately predict moving objects in a 3D environment over long time-spans. a)** Visualization of rollout (after 50 observation frames). FloWM stays consistent until the final frame, while the baselines and ablations hallucinate object position and color. **b)** Corresponding averaged MSE per timestep increases over time for baselines.

**Results.** On MNIST World, we train and evaluate the Simple Recurrent FloWM introduced in Section 3.2, which includes velocity channels (VC) and self-motion equivariance (SME). We also include ablations FloWM (no VC), FloWM (no VC, no SME), and the diffusion baselines mentioned above. We note here that FloWM (no VC, no SME) is just a simple convolutional RNN. More training and model details are available in §H and §K.2. In Table 1 we report multiple quality metrics (MSE, PSNR, & SSIM (Wang et al., 2004)) with respect to the ground truth averaged over the predicted frames for each model. Example rollouts are shown in Fig. 5(a).

Predictions from FloWM remain consistent with ground truth for 150 timesteps past the observation window, *well beyond its training prediction horizon of 20 timesteps*, while FloWM (no VC, no SME) fails, and FloWM (no VC) diverges over time. Length extrapolation errors vs. time are plotted in Fig. 5(b). We further find that models combining SME and VC require orders of magnitude less training steps to converge compared with those without these priors, shown in Fig. 5(c). We see the DFoT's predictions quickly diverge from the ground truth, instead generating plausible digit-like artifacts. Such 'hallucinations' make the model's MSE error worse than the simple all-black baseline. The DFoT-SSM model's predictions show the digits slowly fading to black as it loses track of the digits through the decaying SSM memory. Through additional results in §F, we explore how the DFoT model can sometimes handle partial observability, object dynamics, and self-motion individually, but not in any combination.

### 4.3. 3D Dynamic Block World Benchmark

**Dataset.** Reasoning about the dynamics of the 3D world from 2D image observations requires approximating unprojection of egocentric views to a world-centric representation. To validate FloWM on this more difficult setting, we further introduce a simple 3D dataset, built in the Miniworld environment (Chevalier-Boisvert et al., 2023). An agent is spawned in a random position in a square room, along

with colored blocks initialized with random positions and velocities. At each timestep, the blocks evolve according to their velocities, and the agent takes one of four discrete actions: turn left, turn right, move straight, or do nothing. The blocks bounce when encountering a wall, making the task of modeling dynamics out of view significantly harder. The data-generating agent follows a biased random exploration strategy, sometimes pausing to observe the room's dynamics from next to the wall. During training, the model is given 50 observation frames as context and must predict the next 90 observation frames, given the agent's actions. Experiment details are in §G, and additional results on textured and static variants of Block World are in §E.

**Results.** On the 3D Dynamic Block World dataset, we compare our Transformer-Based FloWM from Section 3.2 and Fig. 4 with the diffusion and RSSM baselines, also including the ablations FloWM (no VC) and FloWM (no VC, no SME). We report the metrics on rollouts of 70 and 210 prediction frames, given 70 frames of context in Table 2[1]. Example rollouts against Diffusion baselines are visualized in Fig. 6 and all baseline model qualitative visualizations are available on the website here.

Similar to the 2D experiments, we observe that FloWM's predictions remain consistent for as many as 210 frames of future prediction, while the baselines diverge. The average prediction error per frame is plotted in Figure 6, demonstrating FloWM's consistent rollouts through long horizons. Perceptually, the DFoT and SSM models frequently hallucinate new objects and forget old ones, while the RSSM model degrades to a blurry average of many overlapping blocks, even in early prediction frames. In Appendix §E, we see the same performance gap holds on more visually complex variants of the Block World dataset, where the set of textures and shapes are randomly assigned for each video, rather than just color. This supports the applicability

---

[1]We report results on 70 frames of context to match the DFoT-SSM training requirements better, and find either 50 or 70 frames of context produce similar results; further explanation is in §K.4.

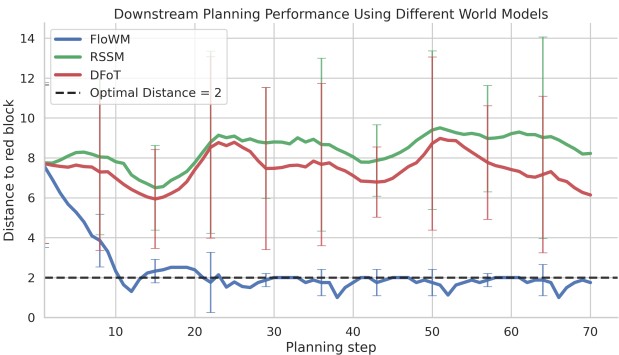

*Figure 7.* **Downstream Planning on Block World.** Distance to the red block using a simple MPC planning algorithm for different learned world models. The optimal distance for the reward function is at a distance of 2 away.

of FloWM to more realistic datasets, and reinforces that the bottleneck of existing world models is less so modeling the visual complexity of the environment, but rather is the challenge of predicting *consistent environment dynamics*.

**Downstream Planning.** To further evaluate the effectiveness of the learned world models for downstream planning, we present results on a simple task in the Block World environment. To enable training-free evaluation, we use a reward function that maximizes the amount of red pixels on the screen called "Find the red block", analogous to being as close to the red block as possible without being on top of it. The max reward is at distance 2, and there is only one red block in each episode. We use exhaustive search in an MPC framework with a search horizon of 3 and a rollout length of 70 after observing 70 context frames.

In total the experiment is run across 8 episodes, and requires no additional trained parameters. As presented in Figure 7 and Table 3, FloWM quickly finds its way to the red block, while the baselines often hallucinate a red block and follow that action, resulting in poor performance. Though this task is relatively simple, we believe it demonstrates the downstream utility of being able to properly predict dynamics in partially observed settings, and the harm of world model hallucinations rampant in the baselines. More experiment and algorithm details are available in §D.

**Learned Equivariant Representation.** To investigate the structure of our learned hidden map more precisely, we train a small probe network to decode block locations from the internal state of each world model. In Figure 8 we plot the decoded positions over time of two blocks (blue & green) from a representative test sequence. We see that the FloWM (left) perfectly predicts the ground truth linear trajectories, while the DFoT decodes a jagged inconsistent path. Ultimately, over the entire test set, we find the FloWM probe is able to decode block position with **96**% accuracy while the DFoT & DFoT-SSM probes decode less than 1% of timesteps correctly.

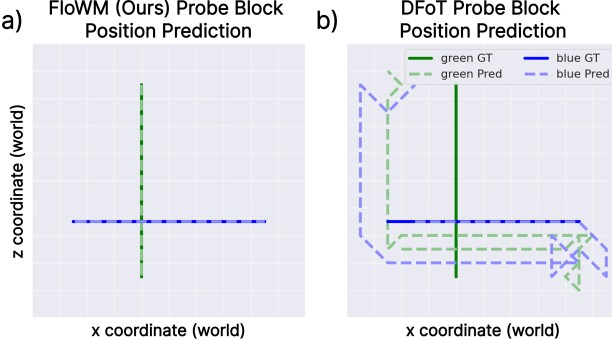

*Figure 8.* **Ground truth block positions and trained probe predictions** visualized for two of six blocks in a Dynamic Block World rollout (top down view). **a)** Probes on FloWM's learned latent space easily recover the ground truth locations. **b)** Baseline (DFoT) predictions result in hallucinated block locations.

Finally, we then leverage these probes to quantitatively test one core assumption made in constructing the 3D FloWM model – that the FloWM ViT Encoder learns to become equivariant to object motion through training. Precisely, we compute an approximate 'equivariance error' as the difference between the decoded position of the block at time $t + 1$, and the decoded position of the block at time $t$ plus the ground truth offset of the block from $t \rightarrow t + 1$. This can be seen as an approximation to the degree to which Equation 5 holds, i.e. $Err = \text{probe}(E_\theta \left[ g \cdot f_t; \; g \cdot h_t \right]) - g \cdot \text{probe}(E_\theta \left[ f_t; h_t \right])$.

We find that before training, the FloWM has an equivariance error of 6.96 (in L2 distance) meaning the original predictions are off by roughly 5 units in both the spatial directions ($\sqrt{5^2 + 5^2}$). After training, we find the error is reduced to **0.22**, an error of only 0.16 units in each dimension. Comparatively, the DFoT model achieves an equivariance error of 2.36. All probe and experiment details are in §C.

## 5. Related Work

**Generative World Modeling with Memory.** Augmenting diffusion-based world models with memory is an active field of research; however, existing work to date has mainly focused on static scenes and incorporate action conditioning through implicit token-based action embeddings. These approaches have thus far lacked a structured mechanism to jointly integrate agent actions and world dynamics within a unified memory representation, which we demonstrate are the key aspects contributing to FloWM's success. The baseline DFoT-SSM model incorporates a recurrent SSM backbone for long horizon prediction in static environments; however, the SSM serves primarily to remember viewpoint-dependent observations, similar to extending the self-attention context window via a linear attention approximation, and does not have any explicit computations to predict the state of the world out of the view of the agent (Po et al., 2025; Dao & Gu, 2024). Another recent model, WORLDMEM, places past image observations in a memory

*Table 1.* **Rollout metrics on 2D MNIST World.** Models are evaluated by conditioning on 50 context frames, then generating 20 (training length) and 150 (length generalization) future frames.

| Model | MSE ↓ | | PSNR ↑ | | SSIM ↑ | |
|---|---|---|---|---|---|---|
| | 20 | 150 | 20 | 150 | 20 | 150 |
| **FloWM (Ours)** | **0.0005** | **0.0018** | **32.99** | **27.56** | **0.9900** | **0.9813** |
| (no VC) | 0.0041 | 0.0334 | 23.83 | 14.77 | 0.9576 | 0.7729 |
| (no SME) | 0.1234 | 0.1317 | 9.088 | 8.805 | 0.0366 | 0.0127 |
| (no SME, no VC) | 0.1233 | 0.1333 | 9.091 | 8.751 | 0.0374 | 0.0146 |
| (+ action-concat) | 0.1125 | 0.1359 | 9.491 | 8.669 | 0.0623 | 0.0149 |
| DFoT | 0.1448 | 0.2111 | 8.394 | 6.755 | 0.4045 | 0.2434 |
| DFoT-SSM | 0.1277 | 0.1688 | 8.940 | 7.726 | 0.4550 | 0.3146 |

*Table 2.* **Rollout metrics on 3D Dynamic Block World.** Given 70 context frames, models are evaluated on generation of 70 (training objective) and 210 (length extrapolation) future prediction frames.

| Model | MSE ↓ | | PSNR ↑ | | SSIM ↑ | |
|---|---|---|---|---|---|---|
| | 70 | 210 | 70 | 210 | 70 | 210 |
| **FloWM (Ours)** | **0.000603** | **0.001539** | **32.19** | **28.13** | **0.9673** | **0.9525** |
| (no VC) | 0.007615 | 0.009614 | 21.18 | 20.17 | 0.9045 | 0.8935 |
| (no SME, no VC) | 0.009579 | 0.012625 | 20.19 | 18.99 | 0.8782 | 0.8631 |
| DFoT | 0.011759 | 0.021684 | 19.30 | 16.64 | 0.9377 | 0.8885 |
| DFoT-SSM | 0.022616 | 0.022570 | 16.46 | 16.46 | 0.8877 | 0.8879 |
| DreamerV3 RSSM | 0.01636 | 0.01647 | 17.86 | 17.83 | 0.8799 | 0.8782 |

*Table 3.* **Planning Metrics on Block World.** Agent distance from the optimal location for the "Find the red block" task.

| Model | Mean |Dist - 2| | Final |Dist - 2| |
|---|---|---|
| **FloWM (Ours)** | **0.727** | **0.250** |
| DreamerV3 RSSM | 6.449 | 6.220 |
| DFoT | 5.571 | 4.138 |

bank for later retrieval based on the camera position (Xiao et al., 2025). This mechanism cannot handle dynamic environments and relies on self-attention to integrate self-motion information, leading to long-term consistency errors. Recent work from (Zhou et al., 2026) maintains a 3D voxel map of the environment, later retrieved to condition diffusion generation. However, their method relies on depth unprojection, the voxels are updated via max pooling instead of a more flexible recurrence relation, and are prohibitively expensive for large hidden state sizes. We refer readers to §B for a complete overview of recent advances in world modeling, including more detail on memory-augmented approaches.

**Equivariant World Modeling.** Most related to our proposed FloWM are works that propose to utilize a similarly structured 'map' memory. Neural Map (Parisotto & Salakhutdinov, 2017) introduced a spatially organized 2D memory that stores observations at estimated agent coordinates that are shifted precisely according to the agent's actions, yielding an effectively equivariant 'allocentric' latent map. A variant of the model can in fact be seen as a special case of our FloWM without velocity channels. Similarly, EgoMap (Beeching et al., 2020) leveraged inverse perspective transformations to map from observations in 3D environments to a top-down egocentric map. Our work can be seen to formalize these early models through our general group theoretic framework, allowing us to extend the action space beyond just spatial translation to any Lie group and any world space. Furthermore, these past works are not predictive world models of the observation space. Other work discusses equivariant world modeling, but is less precisely related to our own: Van der Pol et al. (2020) & Delliaux et al. (2025) build equivariant policy and value networks for reinforcement learning, but with respect to the symmetries of the environment (such as static rotations) rather than dynamic motion. Recent work (Park et al., 2022; Ghaemi et al., 2026) proposes to approach the goal of equivariant world modeling in a more approximate manner by conditioning or encouraging equivariance through auxiliary training losses, rather than our approach which directly proposes a flow equivariant representation space.

## 6. Discussion

In this paper, we have established how the principle of flow equivariance both demands and guides the construction of structured memory for world models in partially observed dynamic environments. Flow equivariant world models are able to represent motion in a structured symmetric manner, permitting faster learning, lower error, fewer hallucinations, and more stable rollouts far beyond the training length. We believe this work lays the theoretical groundwork along with empirical validation to support novel symmetry-structured approaches for efficient and effective world modeling.

**Limitations & Future Directions.** We evaluate FloWM in controlled settings with rigid-body geometric actions and known action parameterizations to isolate the contribution of flow-equivariant memory under partial observability. An important next step is extending our framework to more realistic and open-world datasets. For example, a more powerful diffusion decoder and an adaptively sized latent map able to be maintained for infinite horizons could bring the benefits of FloWM's framework to larger scale self-driving or robotics datasets (Wu et al., 2023). Additionally, our framework could be extended to richer action spaces, such as semantic actions or smooth deformations, potentially via mechanisms that directly learn these transformations from data (Yang et al., 2024a). Similarly, future work may extend FloWM beyond the current discrete velocity sets $V$ to continuous families; however prior empirical and theoretical results suggest that even discrete approximations to continuous groups are beneficial in practice (Kuipers & Bekkers, 2023; Petrache & Trivedi, 2023). Finally, FloWM's latent memory naturally supports non-generative world-model objectives and could be combined symbiotically with objectives such as JEPA for improved latent representation learning for world modeling (Assran et al., 2025).

## Acknowledgments

The authors would like to thank Domas Buracas, Zhiyi Li, Nicklas Hansen, Nathan Cloos, Christian Shewmake, and William Chung for their helpful discussion and feedback about early versions of this work. We would like to especially acknowledge Kirill Dubovitskiy for helping with implementation of an early version of the partially observed dynamic dataset. This work has been made possible in part by a gift from the Chan Zuckerberg Initiative Foundation to establish the Kempner Institute for the Study of Natural and Artificial Intelligence at Harvard University. We would also like to gratefully acknowledge support from PickleRobotics and Amazon AGI Labs.

## Impact Statement

This paper presents work whose goal is to advance the field of Machine Learning. There are many potential societal consequences of our work, none which we feel must be specifically highlighted here.

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

## Appendix Contents

# A. Generalized Flow Equivariance Proof

In this section, we prove by induction that the generalized Flow Equivariant Recurrence Relation of Equation 4 is indeed flow equivariant, following the proof technique of (Keller, 2026).

*Problem statement:* We wish to show that for the recurrence defined as follows:

$$h_{t+1}(\nu) = \psi_1(\nu) \cdot U_\theta \big[ h_t(\nu); \ E_\theta \left[ f_t, h_t \right](\nu) \big], \tag{9}$$

if we assume that:

1. The encoder satisfies the 'trivial lift' condition to the velocity field: $E_\theta \left[ f_t; h_t \right](\nu) = E_\theta \left[ f_t; h_t \right](\hat{\nu}) \ \forall \nu, \hat{\nu} \in \mathfrak{g}$

2. The encoder and update operation are both group equivariant with respect to their arguments: $E_\theta \left[ g \cdot f_t; \ g \cdot h_t \right] = g \cdot E_\theta \left[ f_t; h_t \right] \quad \& \quad U_\theta \left[ g \cdot h_t; \ g \cdot o_t \right] = g \cdot U_\theta \left[ h_t; \ o_t \right]$

3. The hidden state is initialized to be constant along the flow dimension and invariant to the flow action: $h_0(\nu) = h_0(\nu') \ \forall \nu', \nu \in V$ and $\psi_1(\nu) \cdot h_0(\nu) = h_0(\nu) \ \forall \nu \in V$,

then, the following flow-equivariance commutation relation holds:

$$h_t[\psi(\hat{\nu}) \cdot f](\nu) = \psi_t(\hat{\nu}) \cdot h_t[f](\nu - \hat{\nu}) \ \forall t. \tag{10}$$

*Definitions:* We first recall some definitions from the main text. We define $f = \{f_t\}_{t=0}^T$ as the input signal, such as a sequence of $T$ images. The signal at each timestep is defined as $f_t : X \to \mathbb{R}^K$, a mapping from the input space $X$ to a $K$-channel output; for this work we consider the input space to be the group $G$, so $X = G$. Furthermore, in all settings in this work, $G$ refers to the 2D translation group, either indexing pixel coordinates (MNIST), or 2D latent map coordinates (Block World) referred to as $(x, y)$ in the main text.

The output $h$ of a sequence-to-sequence model $\Phi$ are defined as $h = \{h_t\}_{t=0}^T$, where the hidden state at each timestep defines a map $h_t : X' \to \mathbb{R}^{K'}$; the input space of the recurrent state $X'$ is mapped to a $K'$-channel output. In this work we consider hidden states with 'velocity channels', and with the input space otherwise defined over group $G$, so the input space is $X' = V \times G$. Elements within $G$ can be thought of as spatial coordinates of the hidden state. The group $G$ here does not have to be the same as the one defined for the input space of $f$, but we assume that here for clarity. Notation is sometimes suppressed in the main text and here for brevity; specifically, the maps are defined as $h_t : V \times G \to \mathbb{R}^{K'}$, $h_t(\nu) : G \to \mathbb{R}^{K'}$, and $h_t(\nu, g) \in \mathbb{R}^{K'}$

The flow $\psi_t(\nu)$ represents the flow after following the vector field $\nu$ for $t$ timesteps, represents group element $g_t$. We denote the sequence of integrated flows as $\psi(\nu) = \{\psi_t(\nu)\}_{t=0}^T$.

*Theorem* (The Generalized Flow Equivariant Recurrence Relation of Eqn. 9 is Flow Equivariant). Let $h[f] = \{h_t[f]\}_{t=0}^T$ be the output of the generalized flow equivariant recurrence relation as defined in Equation 9 when given an input $f$. The hidden state initialization is invariant to the group action and constant in the flow dimension, i.e. $h_0(\nu, g) = h_0(\nu', g) \ \forall \nu', \nu \in V$ and $\psi_1(\nu) \cdot h_0(\nu, g) = h_0(\nu, g) \ \forall \nu \in V, g \in G$. Then, $h[f]$ is flow equivariant with the following representation of the action of the flow in the output space for $t \geq 1$:

$$(\psi(\hat{\nu}) \cdot h[f])_t(\nu, g) = h_t[f](\nu - \hat{\nu}, \psi_t(\hat{\nu})^{-1} \cdot g) \tag{11}$$

We note for the sake of completeness, that this then implies the following equivariance relation:

$$h_t[\psi(\hat{\nu}) \cdot f](\nu, g) = h_t[f](\nu - \hat{\nu}, \psi_t(\hat{\nu})^{-1} \cdot g) = \psi_t(\hat{\nu}) \cdot h_t[f](\nu - \hat{\nu}, g) \tag{12}$$

We see that, different from the work of (Keller, 2026), since the generalized recurrence relation in Eqn. 9 applies the flow update after the input has been combined with the hidden state (i.e. outside the $U_\theta$ operator), the commutation relation in Eqn. 10 now has a $t$ index on $\psi_t$, instead of $t - 1$ as written in Equation 3.

*Proof.* (Theorem, Generalized Flow Equivariance)

Base Case: The base case is trivially true from the initial condition:

$$h_0[\psi(\hat{\nu}) \cdot f_{<0}](\nu, g) = h_0[f_{<0}](\nu, g) \quad \text{(by initial cond. being independent of input)} \tag{13}$$

$$= h_0[f_{<0}](\nu - \hat{\nu}, \psi_t(\hat{\nu})^{-1} \cdot g) \quad \text{(by constant init.)} \tag{14}$$

Inductive Step: Assuming $h_t[\psi(\hat{\nu}) \cdot f](\nu, g) = \psi_t(\hat{\nu}) \cdot h_t[f](\nu - \hat{\nu}, g) \, \forall \nu \in V, g \in G$, for some $t \geq 0$, we wish to prove this also holds for $t + 1$:

Using the Generalized Flow Recurrence (Eqn. 9) on the transformed input, we get:

$$h_{t+1}[\psi(\hat{\nu}) \cdot f](\nu, g) = \psi_1(\nu) \cdot \mathrm{U}_\theta \Big( h_t[\psi(\hat{\nu}) \cdot f](\nu, g) \; ; \; \mathrm{E}_\theta[(\psi_t(\hat{\nu}) \cdot f_t), \, h_t[\psi(\hat{\nu}) \cdot f]](\nu) \Big) \tag{15}$$

$$\text{(by inductive hyp.)} \quad = \psi_1(\nu) \cdot \mathrm{U}_\theta \Big( \psi_t(\hat{\nu}) \cdot h_t[f](\nu - \hat{\nu}, g) \; ; \; \mathrm{E}_\theta[(\psi_t(\hat{\nu}) \cdot f_t), \, \psi(\hat{\nu})_t \cdot h_t[f]](\nu) \Big) \tag{16}$$

$$\text{(trivial lift in } \nu) \quad = \psi_1(\nu) \cdot \mathrm{U}_\theta \Big( \psi_t(\hat{\nu}) \cdot h_t[f](\nu - \hat{\nu}, g) \; ; \; \mathrm{E}_\theta[(\psi_t(\hat{\nu}) \cdot f_t), \, \psi(\hat{\nu})_t \cdot h_t[f]](\nu - \hat{\nu}) \Big) \tag{17}$$

$$\text{(equivariance of } E_\theta) \quad = \psi_1(\nu) \cdot \mathrm{U}_\theta \Big( \psi_t(\hat{\nu}) \cdot h_t[f](\nu - \hat{\nu}, g) \; ; \; \psi_t(\hat{\nu}) \cdot \mathrm{E}_\theta[f_t, \, h_t[f]](\nu - \hat{\nu}) \Big) \tag{18}$$

$$\text{(equivariance of } U_\theta) \quad = \psi_1(\nu) \cdot \psi_t(\hat{\nu}) \cdot \mathrm{U}_\theta \Big( h_t[f](\nu - \hat{\nu}, g) \; ; \; \mathrm{E}_\theta[f_t, \, h_t[f]](\nu - \hat{\nu}) \Big) \tag{19}$$

$$\text{(flow composition)} \quad = \psi_{t+1}(\hat{\nu}) \cdot \psi_1(\nu - \hat{\nu}) \cdot \mathrm{U}_\theta \Big( h_t[f](\nu - \hat{\nu}, g) \; ; \; \mathrm{E}_\theta[f_t, \, h_t[f]](\nu - \hat{\nu}) \Big) \tag{20}$$

$$\text{(by Eqn. 9)} \quad = \psi_{t+1}(\hat{\nu}) \cdot h_{t+1}[f](\nu - \hat{\nu}, g) \tag{21}$$

$$= h_{t+1}[f]\big(\nu - \hat{\nu}, \, \psi_{t+1}(\hat{\nu})^{-1} \cdot g\big). \tag{22}$$

Thus, assuming the inductive hypothesis for time $t$ implies the desired relation at time $t + 1$; together with the base case this completes the induction and proves the Theorem. $\qquad\square$

## B. Additional Related Work

**World Models**  World modeling is commonly framed as learning to predict how an environment evolves over time, conditioned on an agent's actions, enabling "imagination" for data-efficient policy learning and model-based planning (Ha & Schmidhuber, 2018). Recent progress in large scale generative modeling has extended this paradigm to video prediction, yielding generative world models that directly predict future observations in pixel or latent video space (Ball et al., 2025; Agarwal et al., 2025; Xiang et al., 2024; He et al., 2025; Guo et al., 2025; Xiang et al., 2025). When trained well, these models capture both the stochasticity and multimodality of environment dynamics, and can be queried to simulate counterfactual futures under different action sequences for various applications like planning (Yang et al., 2024b; Du et al., 2024; Hou et al., 2026). Our work also falls in this category; the model is trained to predict an agent's visual observations. However, we focus on the regime where accurate prediction requires persistent memory of the world state across long horizons and changing viewpoints.

A dominant current approach to generative world modeling uses transformer backbones trained with diffusion-style objectives, extending to long rollouts via sliding window inference (Chen et al., 2024; Feng et al., 2024; Yin et al., 2025; Huang et al., 2026). In particular, History-Guided Diffusion Forcing, extends previous video diffusion approaches, which denoised a video of a predefined length with bespoke history conditioning strategies, to enable autoregressive video rollouts. The method does this by denoising multiple frames jointly where each frame can have a different noise level, and can leverage self-attention over recent context (with a noise level of zero) within a unified attention window to improve short-range temporal coherence (represented as the DFoT baseline we benchmark against) (Song et al., 2025). However, in partially observed settings, long horizon prediction requires retrieving and updating information that may have been observed far in the past (e.g. objects that moved out of view). Under the common sliding window attention regime, context is necessarily truncated as rollouts grow, so information that leaves the window becomes inaccessible to the model to generate new frames. The model must remain consistent with past observations while simultaneously reflecting state changes that occurred out of view, yet neither requirement is naturally supported once the relevant evidence is outside the attention window (as visualized in Figure 2). Consequently, despite strong perceptual quality, windowed diffusion transformer simulators struggle to maintain globally consistent, stateful predictions over long horizons, especially in scenes that have out of view dynamics.

Another line of work on generative world models uses a non-diffusion reconstruction objective, such as Dreamer V3 (Hafner et al., 2023). These works use a recurrent state to maintain consistency, and suffer from both poor visual quality (diffusion objectives are typically better at generative modeling over signals like image frames), and a lack of memory for long horizon prediction, as we explored through our Dreamer V3 RSSM baseline.

In contrast, many non-generative world models learn compact latent dynamics tailored for control rather than reconstructing future observations. Representative approaches such as TDMPC2 and V-JEPA style predictive learning maintain recurrent latent state and optimize objectives that encourage predictability and useful abstractions for downstream decision making (Hansen et al., 2024; LeCun & Courant, 2022; Assran et al., 2025; Hansen et al., 2026). These methods can be highly effective for planning in continuous control, but they typically do not aim to produce observation space rollouts, and their benchmarks and objectives often place less emphasis on long horizon, out-of-view state tracking in visually rich, partially observed environments. This gap leads to the question we put forward in this work: how to build an action-consistent world model with a persistent state that can stably represent and evolve both self-motion and external object motion over hundreds of steps, enabling accurate prediction even when the relevant dynamics occur outside the current field of view.

**Memory in generative world models**   Local temporal consistency in generative world models can often be maintained using self attention over recent history (e.g. through the history guided diffusion forcing scheme mentioned in the previous paragraph), but this mechanism degrades once rollouts exceed the model's effective attention window. The comprehensive history of past observations cannot all be kept in context; though there is significant work on long context extensions, such as sparse attention or compression techniques, we believe that more principled solutions are needed to represent the world properly (Zhang et al., 2026a;b). In partially observed environments (which we argue all realistic environments eventually are), long horizon prediction requires a persistent and updatable state that can carry information forward even when relevant evidence is no longer in-context. Tracking objects that move out of view and maintaining their state evolution over time according to previously observed dynamics is an example of this. A growing set of methods therefore augment transformer backbones with explicit memory, yet many existing proposals either (i) primarily address static scene consistency, (ii) store viewpoint-variant observations rather than a world-centric state, or (iii) introduce memory structures whose capacity or cost scales poorly with horizon.

One representative design (and the baseline we chose in our experiments) combines a local attention window for short-range coherence with an auxiliary SSM memory module that compresses past observations (Po et al., 2025; Savov et al., 2026). While such hybrids can stabilize generation over modest horizons, in practice we observed they often learn to memorize camera-conditioned appearance rather than maintaining a canonical scene state. This viewpoint-variant storage becomes brittle under viewpoint changes and especially under dynamics that evolve out of view, since the memory must simultaneously (a) retain evidence from past viewpoints and (b) support correct state updates when the scene changes without direct observation. Moreover, when the auxiliary module has limited functional capacity (e.g. linear state updates), it can further constrain the complexity of dynamics and viewpoint variation that can be retained over long horizons. The original paper focuses on static scenes, and the challenge that more realistic dynamic scenes pose to such methods is established in this work.

Other memory strategies trade compression for retrieval. For instance, database-style approaches such as WORLDMEM store past frames (or features) and retrieve relevant context based on field-of-view overlap (Xiao et al., 2025). While retrieval can recover visually similar evidence, storing raw viewpoints can be inadequate for dynamic environments where the correct future state may differ from any previously observed frame. Furthermore, the memory footprint typically grows with the number of observations unless aggressive pruning or summarization is introduced.

A third class maintains explicit geometric memory, such as pointcloud or voxel-based representations. These representations can promote viewpoint invariant consistency, but they are prohibitively expensive to maintain even at a modest scale scale due to the cubic cost of maintaining a 3D voxel grid of features for a scene (Zhou et al., 2026). Further, a recent method that separates dynamic and static components only remembers the static portion through its pointcloud memory; the dynamic portion is generated anew, conditioned on local frames, so the memory is indeed not meant to handle or predict consistent out-of-view dynamics (Wu et al., 2026). More generally, many existing 3D memory methods are designed for view synthesis with largely static memory, making it challenging to represent and update long horizon dynamics under partial observability. Together, these limitations motivate memory mechanisms that are explicitly world-centric, consistent with action conditioning, and dynamically updatable over long horizons.

An additional intuition figure for partially observable dynamic world modeling in 2D environments (such as our MNIST World benchmark) is available in Figure 9.

**Novel View Synthesis and 3D Priors**   A separate line of work in novel view synthesis and 3D reconstruction, such as NeRFs and 3D Gaussian Splatting, focuses on building explicit scene representations that enable photorealistic rendering from new camera poses (Mildenhall et al., 2021; Kerbl et al., 2023). These methods provide strong geometric and multi-view priors, often yielding impressive viewpoint consistency because rendering is mediated by an underlying 3D representation rather than purely by autoregressive image or video modeling. Extensions to dynamic settings (e.g. "4D" variants that model

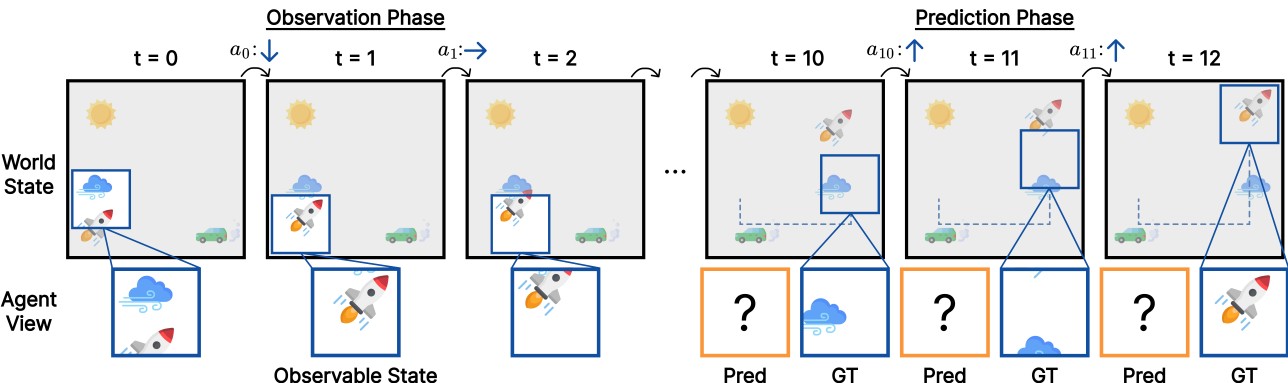

*Figure 9.* **Partially observable dynamic world modeling in 2D environments.** Grayed out areas are not visible to the agent at time $t$. The agent moves its view each timestep via action $a_t$ and must predict future states after an observation phase, conditioned on its own future action sequence.

time varying geometry or appearance) further support rerendering observed motions from novel viewpoints (Pumarola et al., 2020; Liu et al., 2025; Wu et al., 2024). However, these approaches are typically formulated as reconstruction or retargeted rendering problems conditioned on observed frames, rather than as action-conditioned future prediction under partial observability. In other words, they are primarily designed to visualize or interpolate dynamics that are present in the existing data, not to predict how an environment will evolve forward in time when key parts of the state may be unobserved and must be remembered and updated. Further, their representations are often not semantic, so they cannot be used to learn generalized prediction models over future states of the world.

**World Model Evaluation on Long Horizon Memory and Dynamics**    Evaluation of generative world models is often centered on perceptual similarity and sample quality using metrics such as PSNR, SSIM, and distributional scores like FVD (Unterthiner et al., 2019). While useful, these metrics can underemphasize the specific failure modes that arise in long horizon, partially observed prediction: a model may remain visually plausible yet drift in state consistency, forget out-of-view objects, hallucinate new objects, or violate deterministic dynamics in ways that are not strongly penalized by short horizon or ambiguity-tolerant benchmarks. In this work we therefore target a more diagnostic regime: deterministic dynamics under controlled partial observability, where the correct future is well defined and errors can be attributed to deficiencies in memory and state tracking rather than to inherent stochasticity. We view deterministic dynamics modeling as a first step in modeling all classes of dynamics properly, since it is a subset of more general stochastic dynamics. Concretely, we contribute a controlled dataset and protocol that determines whether a world model can (i) retain information beyond a local context window, (ii) update its hidden state when dynamics evolve out of view, and (iii) maintain viewpoint-consistent predictions that reflect an underlying world centric state over long horizons.

**Equivariant World Models**    Equivariant models respect the symmetry of their data, ensuring that structured transformations in the input induce predictable changes in the model's internal state (Cohen & Welling, 2016; Bronstein et al., 2021; Papillon et al., 2025). In addition to providing valuable benefits for 3d pointcloud data, visual representation learning, and molecular dynamics simulation (Fuchs et al., 2020; Shewmake et al., 2023; Batzner et al., 2022), this inductive bias has indeed been found to be valuable in prior work on world modeling as well. Specifically, although not explicitly framed as equivariant, one of the most related world modeling architectures to our proposed self-motion equivariance is the Neural Map (Parisotto & Salakhutdinov, 2017). This work introduced a spatially organized 2D memory that stores observations at estimated agent coordinates. The storage location of these observations is shifted precisely according to the agent's actions, yielding an effectively equivariant "allocentric" latent map. In Section 5 of the paper, the authors describe an egocentric version of their model which can in fact be seen as a special case of FloWM, specifically equivalent to the ablation without velocity channels. The authors demonstrate that their allocentric map enables long-term recall and generalization in navigation tasks. In a similar vein, EgoMap (Beeching et al., 2020) built on this by leveraging inverse perspective transformations to map from observations in 3D environments to a top-down egocentric map. This work also explicitly transforms the latent map in an action-conditioned manner, although the transformation is learned with a Spatial Transformer Network, making it only approximately equivariant. Our work can be seen to formalize these early models in the framework of group theory, allowing us to extend the action space beyond just spatial translation to any Lie group and any world space. For example, our framework can theoretically support full 3-dimensional 'neural maps' without problem, following the framework of flow equivariance. Additionally, these prior works do not make predictions in the observation space of the

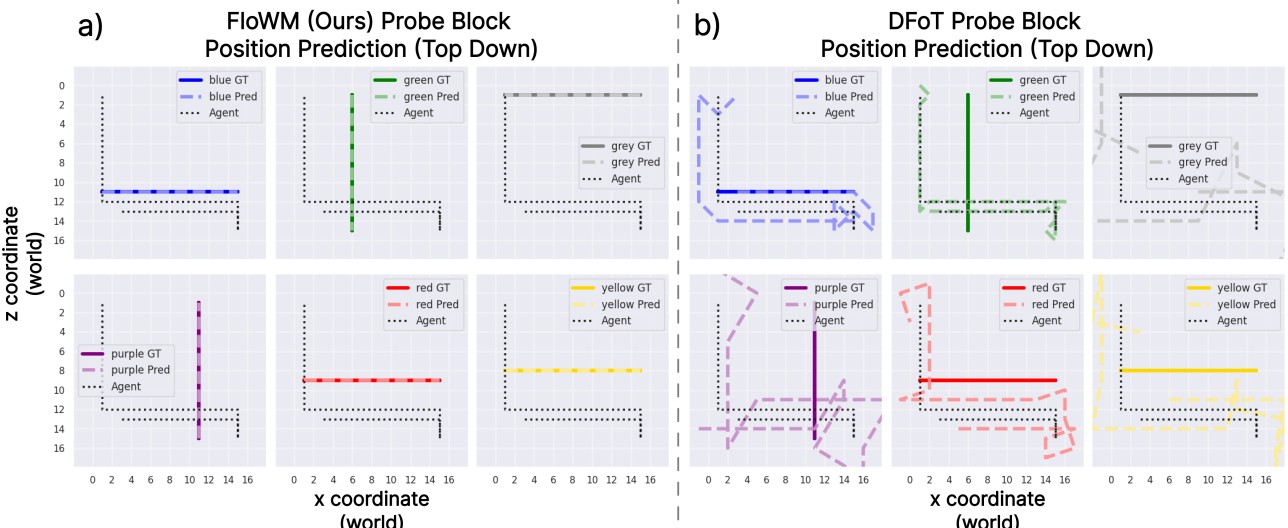

*Figure 10.* **Top Down Probe Visualization for All Colors** on a test set rollout on Dynamic Block World. **a)** Here we visualize the FloWM Probe prediction of the absolute block position. The agent position is ground truth. The predicted block locations remain accurate as the agent moves around, demonstrating the learned flow equivariance of the model. **b)** DFoT predicts sporadic block positions, demonstrating it doesn't maintain a principled representation of the environment and the objects within it.

agent, and do not focus on dynamic scenes. Finally, there are a few other works that discuss equivariant world modeling, but are less precisely related to our own. Specifically, (Van der Pol et al., 2020) was one of the first works to build equivariant policy and value networks for reinforcement learning, but not with respect to motion, instead with respect to the symmetries of the environment (such as static rotations or translations). More recent work (Park et al., 2022; Ghaemi et al., 2026) proposes to approach the goal of building equivariant world models in a more approximate manner by conditioning or encouraging equivariance through training losses, rather than our approach which directly proposes a flow equivariant representation space. Overall, we find all of these approaches to be complementary to our own and are excited for their combined potential.

**Neuroscience.** Excitingly, in the neuroscience literature, there is evidence for predictive processing in the mammalian visual system which is a function of self-motion signals. For example, (Keller et al., 2012) and (Leinweber et al., 2017) have found that responses in visual cortex are strongly modulated by self-motion signals, and mismatch between predicted and experienced stimuli. Similarly, it is known that position coding in the hippocampus through place cells and grid cells forms an equivariant map through phase coding of agent location. Explicitly, the phase of a given place cell's spike shifts *equivariantly* with respect to the agent's forward motion along a linear track. Further computational and theoretical work has demonstrated that grid-like activations emerge automatically through the enforcement of equivariance in cell responses (Dorrell et al., 2023). We believe this work suggests that there may indeed be a biological mechanism for encouraging the visual system to be a Group-equivariant map from stimuli to a type of latent Group-structured space.

# C. Block World Latent Representation Probe Experiments

In this section, we describe the training details and additional results for the representation probe and empirical equivariance experiments introduced in Section 4.3. Broadly, we collect activations from trained models on the Dynamic Block World dataset, then train simple probe models to test whether they can predict the position of blocks from the activations only. This inherits the partial observability, so prediction is evaluated even when the block is out of view of the agent.

## C.1. Probe Experimental Setup

We first run inference on 1024 videos from the Dynamic Block World dataset using the trained FloWM, DFoT, and DFoT-SSM models to collect their activations. For simplicity, in this subset, each environment instantiation has only one block of each color: [red, blue, green, purple, grey, yellow]. This environment setting appears frequently in the training of each of the world models, so it is within the training distribution. Each video matches the 140 frame validation setup, meaning 70 frames are provided as ground truth observations, and the model must predict 70 future frames. Along with the video data, we also collect the ground truth block positions at each timestep, relative to the data-collecting agent. These serve as the training targets for our probe experiments.

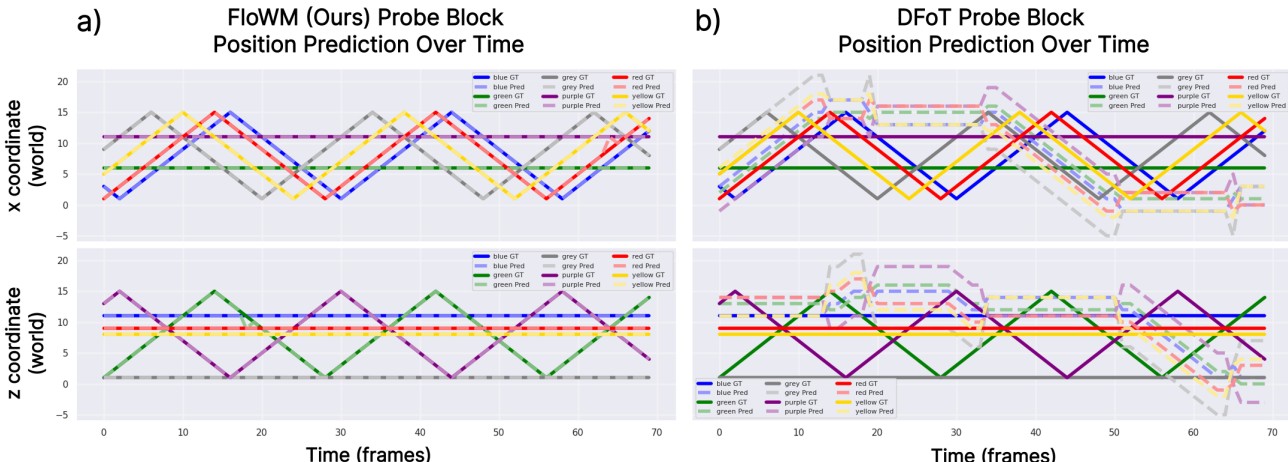

*Figure 11.* **Probe Prediction Through Time** on a test set rollout on Dynamic Block World. **a)** Here we visualize the position prediction for the spatial axes through time. The FloWM model only has a slight error on some of the coordinates, and is able to correct itself. **b)** DFoT's predictions through time do not appear to follow the ground positions closely.

For FloWM, we can directly harvest the hidden state $h_t$ at each prediction timestep $t$. For the baseline models, we instead harvest an intermediate layer activation approximately in the middle: layers 6 and 3 for DFoT and DFoT-SSM respectively (DFoT-SSM has interspersed Transformer and SSM layers, leading to a fewer number of Transformer layers in total when matching parameter counts). Because they are diffusion models, the input passes through the model multiple times for one inference run, corresponding to the number of DDIM sampling steps (see §K.2). We choose to collect the activations from the last pass through the model, when the data is almost clean.

After collecting the activations, we split the dataset into train and test splits by video, retaining $5\%$ for the test set. Then to train the probe model, we treat each timestep as a separate prediction task: given the activation at that timestep $t$, predict the position (relative to the agent) of a particular color of block.

### C.2. Probe Model Details

Due to the difference in the latent representations for FloWM and DFoT, they have slight variations in the probe model architecture, but follow the general same pattern: two layer $3 \times 3$ convolution over the raw activations, flatten, then use a simple 2 layer MLP to predict a distribution over possible locations for the block. DFoT and DFoT-SSM have a spatially organized latent space, but the spatial coordinates are image coordinates; FloWM has a spatial memory as its latent representation. Both are convolved over and then bottlenecked down to a smaller dimension for the flatten operation. FloWM uses a bottleneck dimension of 4; DFoT and DFoT-SSM use a larger bottleneck dimension of 128 to match the parameter count (using 4 for these models did not result in better performance).

Each model class has one probe per block color for simplicity – the overall probe accuracy is reported on the accuracy of the ensemble, and the accuracy per color was roughly the same for each color. Probe models are trained for approximately 50,000 steps or when they saturate validation accuracy, whichever is reached first. The learning rate is 1e-3 and batch size is 64. The model construction results in approximately 7 million parameters for the probe for each color and trains on 1 L40S GPU within 2 hours.

### C.3. Additional Probe Model Results

In addition to the Probe results in Section 4.3, we present additional visualizations of the Probe experiments here. Figure 10 displays for a representative rollout, the top down map including the agent's position, ground truth position, and predicted position for both the FloWM and DFoT baseline probe models. Figure 11 presents the same data with a different visualization, plotting the prediction through time. We can see a small error at one of the timesteps for the FloWM probe, which it is able to recover from, in contrast to the sporadic predictions from the DFoT probe. The quantitative results are presented in the main text in Section 4.3.

## D. Block World Downstream Planning Experiment Details

In this section we describe further details on the Block World downstream planning experiment, "Find the red block". The algorithm is described in Algorithm 1. The world is initialized as described in the standard Block World setup, but always

*Algorithm 1.* Model predictive control with world model rollouts.

```python
# H: planning horizon, N: number of sampled action sequences
def run_episode(env, world_model, H, N):
    # Initialize context frames to predict from
    # context: [context_len, C, H_img, W_img]
    context = env.reset()
    done = False

    while not done:
        # Sample candidate action sequences.
        # actions: [N, H]
        actions = sample_actions(num_candidates=N, horizon=H)

        # Imagine future observations under each action sequence.
        # pred_obs: [N, H, C, H_img, W_img]
        pred_obs = world_model.rollout(context=context, actions=actions)

        # Compute predicted reward for each imagined frame.
        # rewards: [N, H]
        rewards = compute_reward(pred_obs)

        # Score each rollout by total predicted reward.
        # scores: [N]
        scores = rewards.sum(dim=1)

        # Choose the best imagined action sequence.
        best_idx = scores.argmax()

        # Execute only the first action in the real environment.
        best_action = actions[best_idx, 0]
        next_obs, reward, done, info = env.step(best_action)

        # Update real context using the real observation from the environment.
        context = update_context(context=context, action=best_action, obs=next_obs)

    return
```

with 6 blocks to guarantee there is only one of each color: [red, blue, green, purple, grey, yellow]. For the experiments presented in this paper, `sample_actions` is an exhaustive search over a horizon of length 3, where there are 4 possible actions: turn left, turn right, move forward, or do nothing. The function `compute_reward` calculates the number of red pixels on the screen as a proxy for closeness to the red block.

There are many more sophisticated planning algorithms that could be implemented with these world models, but the purpose of this experiment is to just demonstrate the base utility of a world model with dynamic prediction capabilities in partially observed environments. Planning rollout visualizations are available on the website here.

## E. Block World Additional Dataset Details and Results

### E.1. Block World Dataset Details

Here, we describe dataset generation and parameter settings for the Dynamic (presented in the main text), Textured Dynamic, and Static subsets of Block World. The dataset generation parameters are described in Table 4. Each dataset example has a video of shape [num_frames, channels, height, width], where channels is 3 for RGB; and an accompanying list of actions with shape [num_frames], for the discrete actions taken by the agent: [left, right, forward, or do nothing]. Left and right correspond to a 90 degree rotation such that the agent stays aligned with the grid. Each dataset subset contains 10,000 videos for training, and 1,000 videos for validation. The world size describes the number of coordinates in the world able to be occupied by the agent or a block. The agent is spawned in a random location within the world.

The Dynamic Block World dataset subset has blocks with randomized colors chosen from among [blue, green, yellow, red, purple, and gray]. There are no collisions between the blocks, or between the agent and the blocks, but the blocks

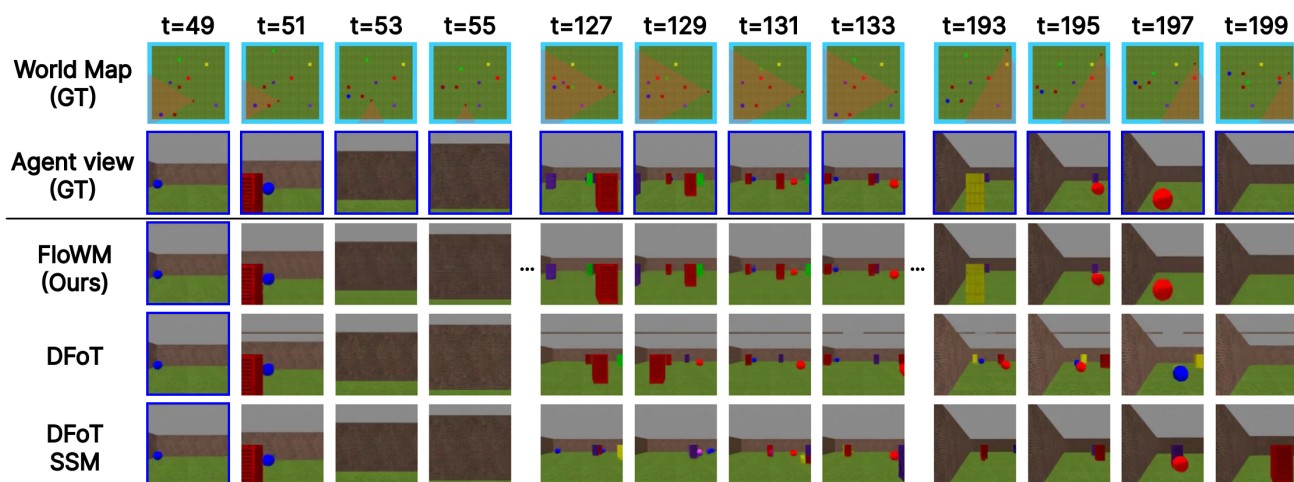

*Figure 12.* **Textured 3D Block World Rollouts.** Here we visualize additional qualitative rollout results on the Textured 3D Block World split. Note the hallucinated objects in both DFoT and DFoT-SSM as time goes on, whereas FloWM remains consistent.

*Table 4.* Generation parameters for Block World dataset subsets.

| Data Subset | World Size | Observation Resolution | # Blocks | Block Velocity Range, x and y |
|---|---|---|---|---|
| Static Block World | 15x15 | 128x128 | 6-10 | 0 |
| Dynamic Block World | 15x15 | 128x128 | 6-10 | -1 to +1 (no diagonals) |
| Textured Dynamic Block World | 15x15 | 128x128 | 6-10 | -1 to +1 (no diagonals) |

bounce off the wall and change direction, making the motion nonlinear, but still deterministic.

The textured dataset has the same dynamics behavior as the Dynamic Block World dataset, but with the following randomizations: (i) textures are randomized for the wall, chosen between [brick, dark wood panel, and wood panel]; (ii) textures are randomized for the floor, chosen between [cardboard, grass, and concrete]; (iii) textures are randomized for the blocks, chosen between [metal grill, airduct grate, cinder blocks, and ceiling tiles]; they remain colored randomly; (iv) 1/3 of the time, the object is a sphere instead of a block. In combination, these all add significant visual complexity and randomness to the environment, which we use to evaluate the applicability of our method to more visually complex datasets, as well as the generalization capability of scenes with more initialization randomness.

The static subset is the same as the dynamic subset, but with the velocity of the objects initialized to 0; the agent's exploration pattern is the same. We used Miniworld for this environment, and would like to thank the authors and contributors of Miniworld for creating a flexible and useful 3D simulator environment for agents (Chevalier-Boisvert et al., 2023).

### E.2. Textured Block World Results

In this section we report additional results on the Textured Dynamic Block World dataset described above. The focus of this work is on the introduction of the flow equivariant framework for modeling partially observable dynamics, rather than the strength of the visual encoder / decoder. However the fact that FloWM can retain its performance improvements over the baselines in this setting suggests it can serve as a step forward for a general framework capable of encoding visually realistic scenes as well.

The results are available in Table 5. Example rollouts are best viewed on the anonymous project website here. Frames of the rollouts are visible in Figure 12. The metric numbers are not easily comparable to the other dataset split due to the different dataset statistics, but the relative distance is still clearly significant. We evaluate with 70 context frames to match the training regime of DFoT-SSM; more details are available in Appendix K.4.

### E.3. Static Block World Results

In this section we report additional results on the static Block World dataset described above. Results are available in Table 6. As with the static MNIST World dataset, in this setting, the default configuration of FloWM with velocity channels only adds noise to the model, since the velocity channels have no external motion to model. Therefore, it is unsurprising that the FloWM (no VC) achieves the best metric scores. Despite the environment being static, DFoT and DFoT-SSM still struggle with keeping consistent with information that may have left their immediate context window. We hypothesize that due to the baseline models' lack of a spatial memory, combined with the randomized number of blocks per environment,

*Table 5.* **Rollout results on Textured 3D Dynamic Block World.** Models are evaluated by generating 70 and 210 future frames conditioned on 70 context frames.

| Model | MSE ↓ | | PSNR ↑ | | SSIM ↑ | |
|---|---|---|---|---|---|---|
| | 70 | 210 | 70 | 210 | 70 | 210 |
| **FloWM (Ours)** | **0.000826** | **0.000926** | **30.83** | **30.33** | **0.9388** | **0.9376** |
| DFoT | 0.005581 | 0.009050 | 22.53 | 20.43 | 0.8960 | 0.8577 |
| DFoT-SSM | 0.009658 | 0.012169 | 20.15 | 19.15 | 0.8581 | 0.8384 |

*Table 6.* **Rollout performance on 3D Static Block World with partial observability.** Models are evaluated by generating 70 or 210 future frames, conditioned on 70 context frames.

| Model | MSE ↓ | | PSNR ↑ | | SSIM ↑ | |
|---|---|---|---|---|---|---|
| | 70 | 210 | 70 | 210 | 70 | 210 |
| **FloWM (Ours)** | 0.000860 | 0.000937 | 30.65 | 30.28 | 0.9675 | 0.9629 |
| (no VC) | **0.000789** | **0.000641** | **31.03** | **31.93** | **0.9800** | **0.9803** |
| (no SME, no VC) | 0.015230 | 0.015816 | 18.17 | 18.01 | 0.8586 | 0.8510 |
| DFoT | 0.011686 | 0.021069 | 19.32 | 16.76 | 0.9378 | 0.8910 |
| DFoT-SSM | 0.004763 | 0.007985 | 23.22 | 20.98 | 0.9743 | 0.9576 |

these models are unable to consistently remember where the blocks are. Here we also evaluate with 70 context frames to match the training regime of DFoT-SSM; more details are available in Appendix K.4,

## F. MNIST World Additional Dataset Details and Results

### F.1. MNIST World Dataset Details

| Data Subset | Self-Motion | Dynamics | Partially Observable |
|---|---|---|---|
| `dynamic_fo_no_sm` | No | Yes | No |
| `dynamic_fo` | Yes | Yes | No |
| `static_po` | Yes | No | Yes |
| `dynamic_po` | Yes | Yes | Yes |

*Table 7.* MNIST world data subsets demonstrating scaling difficulty in self-motion, dynamics, and partial observability.

Here, we describe dataset generation and parameter settings for our ablations on self-motion, dynamics, and partial observability in the MNIST World setting. The subsets are succinctly described in Table 7, and the generation parameters are in Table 8. A subset is described as partially observable if the world size is larger than the window size. We also scale the number of digits by the size of the world. Each dataset example has a video of shape `[num_frames, channels, height, width]`, where channels is 1; and an accompanying actions list of shape `[num_frames, 2]`, for the x and y translation of the agent view at each timestep. The `dynamic_fo_no_sm` subset just has dynamics and is fully observable; the `dynamic_fo` subset has dynamics and is fully observable, but also has self-motion; the `static_po` subset is partially observable, and the agent has self-motion, but the digits do not move; and finally, the `dynamic_po` subset includes partial observability, agent movement, and dynamics. In the main text, we report all results on just the `dynamic_po` subset. For all subsets with dynamics, each digit is given an integer velocity for x and y in the digit velocity range (e.g., -2 to 2). For each dataset subset with self-motion, at each step during the observation and prediction phase, a random integer is chosen in x and y to be the agent's view translation, bounded by the self-motion range (e.g., -10 to 10). For each dataset, objects that move across the boundary reappear on the other side as a circular pad. Each dataset subset contains 180,000 videos in the training set, and 8,000 videos in the validation set. Results on each of these data subsets for FloWM and baseline models are described below.

### F.2. MNIST World Additional Results

Here we report additional results on the MNIST World subsets described above. We evaluate FloWM and FloWM ablations described in Appendix H. We compare to the DFoT and DFoT-SSM baselines as described in Appendix K.4.

The error (MSE) between the predicted future observations (rollout) and the ground truth is plotted for each baseline in Figure 13 as a function of forward prediction timestep (x-axis). Metrics are reported over the first 20 timesteps (the training length) and over the full 150 timesteps (length generalization) in Tables 9, 10, 11. Due to being constructed with a different

| Data Subset | World Size | Window Size | # Digits | Self-motion Range | Digit Velocity Range, x and y |
|---|---|---|---|---|---|
| dynamic_fo_no_sm | 32 | 32 | 3 | 0 | -2 to +2 |
| dynamic_fo | 32 | 32 | 3 | 10 | -2 to +2 |
| static_po | 50 | 32 | 5 | 10 | 0 |
| dynamic_po | 50 | 32 | 5 | 10 | -2 to +2 |

*Table 8.* Generation parameters for MNIST World dataset subsets.

*Table 9.* **Rollout performance on 2D Dynamic MNIST World with full observability and without self-motion, `dynamic_fo_no_sm`.** Models are trained to generate 20 future frames given 50 context frames, and evaluated by generating 20 (training length) and 150 (length generalization) future frames respectively, conditioned on 50 context frames.

| Model | MSE ↓ | | PSNR ↑ | | SSIM ↑ | |
|---|---|---|---|---|---|---|
| | 20 | 150 | 20 | 150 | 20 | 150 |
| **FloWM (Ours)** | 0.00009 | **0.00104** | 40.61 | **29.83** | **0.9982** | 0.9752 |
| (no SME) | **0.00006** | 0.00173 | **41.89** | 27.61 | 0.9981 | **0.9797** |
| (no SME, no VC) | 0.00031 | 0.02845 | 35.05 | 15.46 | 0.9950 | 0.7746 |
| (no VC) | 0.00026 | 0.00925 | 35.87 | 20.34 | 0.9877 | 0.9061 |
| (action concat) | 0.00015 | 0.01349 | 38.36 | 18.70 | 0.9973 | 0.8099 |
| DFoT | 0.02128 | 0.03601 | 16.72 | 14.44 | 0.8961 | 0.8205 |
| DFoT-SSM | 0.01298 | 0.22098 | 18.87 | 6.56 | 0.9399 | 0.2907 |

number of digits, metrics between the data subsets are not necessarily directly comparable.

All models are able to do reasonably well on the simplest fully observable dataset with no self-motion; note here the DFoT is doing latent diffusion, so there is a small amount of error from the decoding step, contributing a baseline MSE of around 0.02, see Appendix K.5 for more details. This setup aligns with the typical setting of world modeling, where the information that the model needs is expected to be in the attention window. The other dataset splits do not follow this assumption, and the results align with expectations about the model's capabilities. The DFoT does relatively better on the static static_po compared to the dynamic_po dataset, due to not having to model dynamics, but the model's outputs still diverge from the ground truth quickly.

For a dataset where the velocity channels are redundant, i.e. static_po, the FloWM (no VC) does slightly better than FloWM. Further note that the FloWM (no VC) is able to have low error on most of the tasks, though with a much higher value than FloWM as errors accumulate due to not having the velocity channels to encode flow equivariantly. Taken together, the ablations suggest that self-motion equivariance is key to solving the problem, and that input flow equivariance via velocity channels helps with exactness and convergence time, with the tradeoff of a larger hidden state activation size.

## G. FloWM Experiment Details: 3D Dynamic Block World

On the 3D Dynamic Block World Dataset, FloWM is built with 6-layer ViT encoders and decoders with 8 attention heads per layer, and an embedding dimension of 256.

### G.1. Recurrence

The hidden state $h_t \in \mathbb{R}^{|V| \times C_{hid} \times H_{world} \times W_{world}}$ has $|V|$ velocity channels (indexed by the elements $\nu \in V$), and $C_{hid} = 256$ hidden state channels (the same as the ViT token embedding dimension). The spatial dimensions of the hidden state are set to two times the world size for each dataset, meaning $H_{world} = W_{world} = 32$, so that the agent can be robust to being spawned anywhere. The hidden state is initialized to all zeros for the first timestep, i.e. $h_0 = \mathbf{0}$.

### G.2. Velocity Channels

On the 3D Block World dataset, we add velocity channels only up to $\pm 1$ in both the X and Y dimensions of the map with no diagonal velocities, and include a zero velocity channel. Thus in total, $|V| = 5$ for the 3D FloWM. Each channel is flowed by its corresponding velocity field, denoted by $\psi_1(\nu) \cdot h_t(\nu)$ for each of the velocities $\nu$, matching the velocity of the blocks in the real environment.

The actions of the agent then induce an additional flow of the hidden state, which we implement via the inverse of the representation of the action $T_{-a_t} = T_{a_t}^{-1}$. In practice, this is implemented by performing a roll operation on the hidden state by exactly 1 element for a forward action, and a $+/-$ 90-degree rotation for left or right actions respectively. Doing so allows the model's representation of the world to stay accurate with respect to the transformation it has just done. Intuitively,

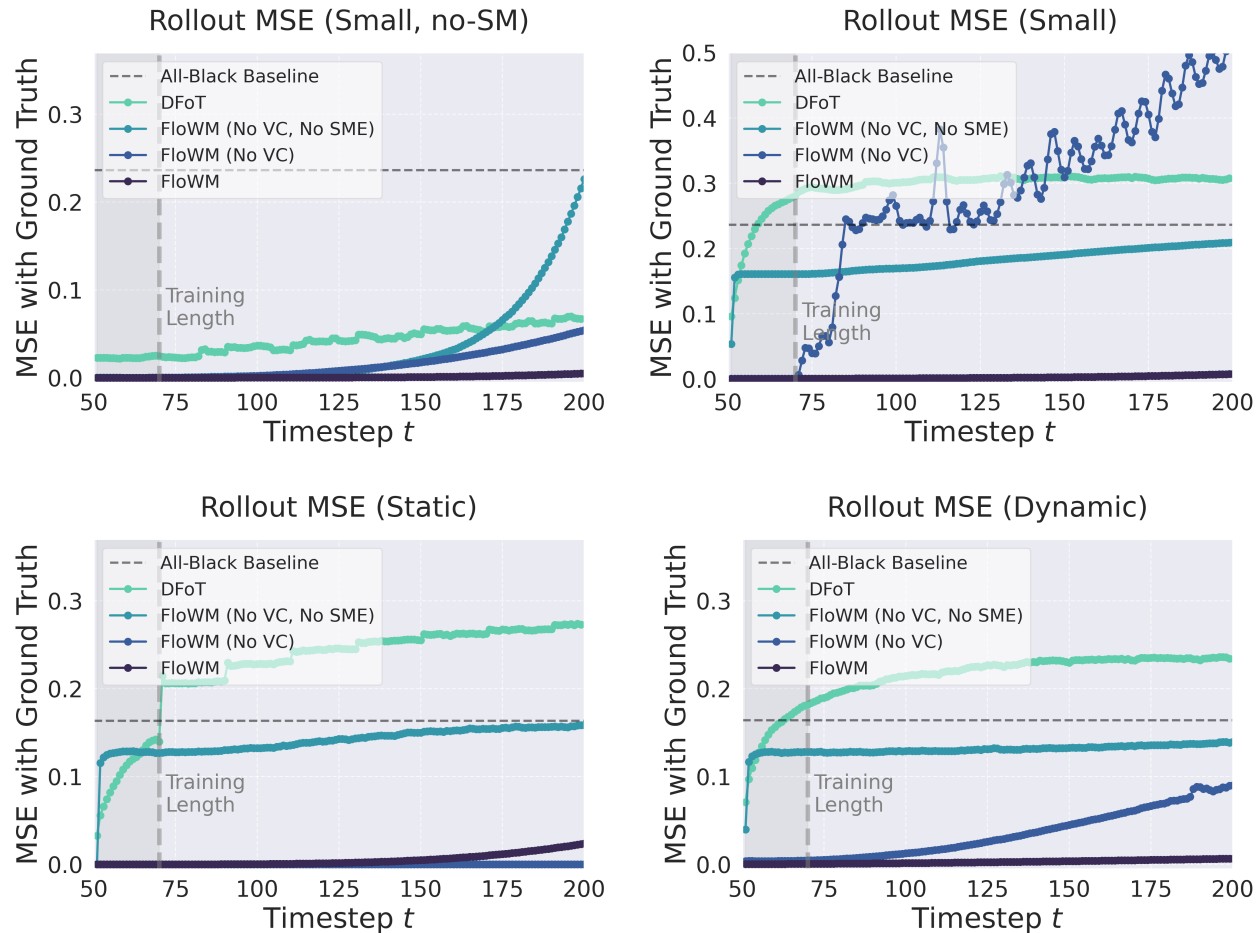

*Figure 13.* Rollout Error (MSE) vs. Forward Prediction Steps for all data subsets of MNIST World. The dynamic subset is replicated from the main text for ease of comparison.

if an agent has a representation of an object in front of it, then turns to the left, the world should shift to the right, opposite of the movement of the turn.

### G.3. Transformer Details

The ViT Encoder takes in both image input tokens and the FoV-selected map latent tokens, and processes them together in the self-attention window. We use a patch size of 16 to patchify the image before a sin-cos absolute position embedding is added (Vaswani et al., 2017). The map latent tokens are also added together with a sin-cos absolute position embedding based on the position in the 2D map. Notably, these two are different embeddings and represent different spatial coordinates: one is of the image patch location, and the other is of the world location relative to the agent. The encoder then produces a separate output for each velocity channel. While this violates the 'trivial-lift' requirement of the equivariance proof, in practice this is simply an overparameterization, and the model has the ability to resort to this trivial lift if it is beneficial. Given the encoder already has to learn to become equivariant in this setting, we find these additional degrees of freedom do not hurt model performance.

The ViT Decoder starts with a copy of a learnable [MASK] token at each patch location in the image, added with position embeddings representing the patch location in pixel space. Then it uses cross attention between the selected tokens in the Field of View of the updated hidden state $\text{FoV}(h_{t+1})$ to make a prediction of the observation at the next timestep, $\hat{f}_{t+1}$.

### G.4. Update Details

The update mechanism in the transformer-based FloWM is implemented as a simple gated combination of the output of the encoder (map token latent positions only), and the previous values of the hidden state $\text{FoV}(h_t)$. Explicitly, matching our

*Table 10.* **Rollout performance on 2D Dynamic MNIST World with full observability, `dynamic_fo`.** Models are trained to generate 20 future frames given 50 context frames, and evaluated by generating 20 (training length) and 150 (length generalization) future frames respectively, conditioned on 50 context frames.

| Model | MSE ↓ | | PSNR ↑ | | SSIM ↑ | |
|---|---|---|---|---|---|---|
| | 20 | 150 | 20 | 150 | 20 | 150 |
| **FloWM (Ours)** | 0.00013 | **0.00136** | 39.01 | **28.65** | 0.9977 | **0.9763** |
| (no SME) | 0.15595 | 0.16167 | 8.07 | 7.91 | 0.0355 | 0.0105 |
| (no SME, no VC) | 0.15597 | 0.18251 | 8.07 | 7.39 | 0.0351 | 0.0141 |
| (no VC) | **0.00010** | 0.26278 | **40.21** | 5.80 | **0.9982** | 0.1718 |
| (action concat) | 0.13425 | 0.15148 | 8.72 | 8.20 | 0.0796 | 0.0294 |
| DFoT | 0.22194 | 0.29699 | 6.54 | 5.27 | 0.2086 | 0.0833 |
| DFoT-SSM | 0.08162 | 0.25271 | 10.88 | 5.97 | 0.6170 | 0.1090 |

*Table 11.* **Rollout performance on 2D Static MNIST World with partial observability, `static_po`.** Models are trained to generate 20 future frames given 50 context frames, and evaluated by generating 20 (training length) and 150 (length generalization) future frames respectively, conditioned on 50 context frames.

| Model | MSE ↓ | | PSNR ↑ | | SSIM ↑ | |
|---|---|---|---|---|---|---|
| | 20 | 150 | 20 | 150 | 20 | 150 |
| **FloWM (Ours)** | 0.00013 | 0.00467 | 38.78 | 23.31 | 0.9970 | 0.8053 |
| (no SME) | 0.12097 | 0.12790 | 9.17 | 8.93 | 0.0460 | 0.0116 |
| (no SME, no VC) | 0.12086 | 0.13046 | 9.18 | 8.85 | 0.0520 | 0.0144 |
| (no VC) | **0.00002** | **0.00004** | **47.63** | **44.49** | **0.9994** | **0.9993** |
| (action concat) | 0.09473 | 0.11860 | 10.24 | 9.26 | 0.1595 | 0.1354 |
| DFoT | 0.10505 | 0.22641 | 9.79 | 6.45 | 0.5199 | 0.2257 |
| DFoT-SSM | 0.05398 | 0.15706 | 12.68 | 8.04 | 0.7274 | 0.3397 |

framework:

$$U_\theta[h_t; o_t]^{(x,y)} = \begin{cases} (1-\alpha) * h_t^{(x,y)} + \alpha * o_t^{(x,y)} & \text{if } x, y \in \text{FoV} \\ h_t^{(x,y)} & \text{otherwise} \end{cases}, \tag{23}$$

where $\alpha = \sigma(\mathbf{W} \operatorname{concat}[h_t^{(x,y)}; o_t^{(x,y)}])$ for some learnable gating weights $\mathbf{W}$. We can see that the update operation $U_\theta$ is indeed equivariant with respect to shifts or rotations of the spatial coordinates of its inputs, satisfying Equation 6 for $U_\theta$.

### G.5. Training Loss Details

To train FloWM, as well as the ablated versions, we provide the model with 50 observation frames as input, and train the model to predict the next 90 observations conditioned on the corresponding future action sequence. Specifically, we minimize the mean squared error (MSE) between the output of the model and the ground truth sequence, averaged over the prediction frames 50 to 139:

$$\mathcal{L}_{MSE} = \frac{1}{90} \sum_{t=50}^{139} ||f_t - \hat{f}_t||_2^2. \tag{24}$$

The models are trained with the Adam optimizer with a learning rate of $1e-4$, a batch size of 16. We use a teacher forcing ratio of $12.5\%$. They are each trained for 150k steps, or until converged. More details are available in Table 12.

## H. FloWM Experiment Details: MNIST World

The Simple Recurrent version of the Flow Equivariant World Model for 2D settings is built as a simple sequence-to-sequence RNN with small CNN encoders/decoders to model MNIST digit features. Full code is available on the project page.

For completeness, we repeat the Simple Recurrent FloWM recurrence relation below:

$$h_{t+1}(\nu) = \psi_1(\nu - a_t) \cdot \sigma\big(\mathcal{W} \star h_t(\nu) + \operatorname{pad}(\mathcal{U} \star f_t)\big). \tag{25}$$

*Table 12.* Block World ViT FloWM Configurations.

| Component | Option | Value |
|---|---|---|
| Training | Learning rate | 1e-4 |
| | Effective batch size | 16 |
| | Training steps | 150k |
| | GPU usage | 2×H100 |
| | Teacher Forcing | 0.125 |
| Model | Hidden channels | 256 |
| | Encoder depth | 6 |
| | Encoder heads | 8 |
| | Decoder dim | 256 |
| | Decoder depth | 6 |
| | Decoder heads | 8 |
| | Patch size | 16 |
| | N Params | 10M |

### H.1. Recurrence

The hidden state $h_t \in \mathbb{R}^{|V| \times C_{hid} \times H_{world} \times W_{world}}$ has $|V|$ velocity channels (indexed by the elements $\nu \in V$), and $C_{hid} = 64$ hidden state channels. The spatial dimensions of the hidden state are set to match the world size for each dataset. For the partially observed world, this means $H_{world} = W_{world} = 50$ (where the window size is set to $32 \times 32$), while for the fully observed world, $H_{world} = W_{world} = 32$. The hidden state is initialized to all zeros for the first timestep, i.e. $h_0 = \mathbf{0}$.

The hidden state is processed between timesteps by a convolutional kernel $\mathcal{W}$. This kernel has the potential to span between velocity channels, and therefore model acceleration or more complex dynamics than static velocities. In this work, since our dataset has no such dynamics (we only have constant object velocities), we safely ignore the inter-velocity convolution terms, and simply set $\mathcal{W}$ to be a $3 \times 3$ convolutional kernel, with 64 input and output channels, circular padding, and no bias. We refer the interested reader to (Keller, 2026) for details on the form of the full flow-equivariant convolution that could be equally used in this model. The hidden state is finally passed through a non-linearity $\sigma$ to complete the update to the next timestep. In this work, for MNIST World, we use a ReLU.

### H.2. Velocity Channels

In this work, for MNIST World, we add velocity channels up to $\pm 2$ in both the X and Y dimensions of the image. Explicitly, $V = \{(-2, -2), (-2, -1), \dots (0, 0) \dots (2, 2)\}$. Thus in total, $|V| = 25$ for the FloWM. Each channel is flowed by its corresponding velocity field (defined by $\psi(\nu)$) at each step. This is denoted by $\psi_1(\nu) \cdot h_t(\nu)$.

The actions of the agent then induce an additional flow of the visual stimulus. In order to be equivariant with respect to this flow in addition to the flows in $V$, we simply additionally flow each hidden state by the corresponding inverse of the action flow $\psi(-a_t)$. In total this gives the combined flow for each flow channel $\psi_1(\nu - a_t)$. In practice, this is implemented by performing a $\mathrm{roll}$ operation on the hidden state by exactly $(\nu - a_t)$ pixels.

### H.3. Encoder

The 'encoder' is simply a single convolutional layer, with $3 \times 3$ kernel $\mathcal{U}$, 1 input channel, and 64 output channels. The convolution uses circular padding, and no bias. The observation at timestep $t$, $f_t$, is thus processed by the encoder $(\mathcal{U} \star f_t)$ yielding the processed observation of the agent. Given this observation is only a partial observation of the full world, we must pad this observation to match the world size, and the size of the hidden state. We denote this operation as 'pad' in the recurrence relation, and simply pad the boundary of the output of the encoder with $0$ to match the world-size (size of the hidden state).

### H.4. Decoder

We learn the parameters of the FloWM by training it to predict future observations from its hidden state and the corresponding sequence of future actions. To compute this prediction, we take a consistent $\mathrm{window\_size}$ crop from the center of the hidden state, corresponding to the same location where the encoder 'writes-in'. We denote this crop $\mathrm{window}(h_{t+1})$. To then enable the model to predict each pixel's velocity independently, we perform a pixel-wise max-pool over the $V$ dimension ('velocity channels') before passing the result to a decoder $g_\theta$. Specifically: $\hat{f}_{t+1} = g_\theta\left(\max_\nu \left(\mathrm{window}(h_{t+1})\right)\right)$. The decoder $g_\theta$ is a simple 2 layer convolutional neural network with $3 \times 3$ convolutional kernels, 64 hidden channels, and a ReLU non-linearity between the layers.

### H.5. Ablation: No Velocity Channels

To construct the ablated version of the FloWM with no velocity channels, we simply set $V = \{(0,0)\}$. Since the original FloWM model simply max-pools over velocity channels, the decoder already only takes a single velocity channel as input, so no other portions of the model need to change. We note that this model is identical to a simple convolutional recurrent neural network with self-motion equivariance. Explicitly:

$$h_{t+1} = \psi_1(-a_t) \cdot \sigma\big(\mathcal{W} \star h_t + \text{pad}(\mathcal{U} \star f_t)\big). \tag{26}$$

### H.6. Ablation: No Self-Motion Equivariance

To construct the ablated version of the FloWM with no self-motion equivariance, we simply remove the term $-a_t$ from the flow of the recurrence relation. Explicitly:

$$h_{t+1}(\nu) = \psi_1(\nu) \cdot \sigma\big(\mathcal{W} \star h_t(\nu) + \text{pad}(\mathcal{U} \star f_t)\big). \tag{27}$$

We note that this is equivalent to the original FERNN model with the addition of the partial-observability modifications (padding the input and windowing the hidden state for readout).

### H.7. Ablation: No Velocity Channels + No Self-Motion Equivariance

To ablate both velocity channels and self-motion equivariance, we reach a simple convolutional RNN:

$$h_{t+1} = \sigma\big(\mathcal{W} \star h_t + \text{pad}(\mathcal{U} \star f_t)\big). \tag{28}$$

### H.8. Ablation: Conv-RNN + Action Concat

We additionally include a version of the model with no velocity channels, no self-motion equivariance, but with action conditioning for both the input and hidden state (denoted 'action-concat'). Specifically, we concatenate the current action to the hidden state vector and the input image as two additional channels (corresponding to the x and y components of the action translation vector), and change the number of input channels for both convolutions correspondingly. Explicitly:

$$h_{t+1} = \sigma\big(\mathcal{W} \star \text{concat}(h_t, a_t) + \text{pad}(\mathcal{U} \star \text{concat}(f_t, a_t))\big). \tag{29}$$

Empirically, we find that this additional conditioning marginally improves the model performance; however, the model is still clearly unable to learn the precise equivariance that the FloWM has built-in.

### H.9. Training Details

To train the FloWM, as well as the ablated versions, we provide the model with 50 observation frames as input, and train the model to predict the next 20 observations conditioned on the corresponding action sequence. Specifically, we minimize the mean squared error (MSE) between the output of the model and the ground truth sequence, averaged over the 20 frames (from frame 50 to 69):

$$\mathcal{L}_{MSE} = \frac{1}{20} \sum_{t=50}^{69} ||f_t - \hat{f}_t||_2^2. \tag{30}$$

The models are trained with the Adam optimizer with a learning rate of $1e-4$, a batch size of 32, and gradient clipping by norm with a value of $1.0$. They are each trained for 50 epochs, or until converged. Some models, such as the FloWM with self-motion equivariance but no velocity channels, took longer than 50 epochs to converge, and thus training was extended to 100 epochs. All 2D FloWM models (and ablations) have roughly 75K trainable parameters.

## I. Compute Resource Comparison

### I.1. Training and Inference Compute Comparisons

To contextualize the computational footprint of FloWM, we report training compute (in EFLOPs), and inference wall-clock time for FloWM and the diffusion-based baselines. Forward, backward, and total training compute are reported in Table 13. FloWM requires roughly 1.7 to 2.6× more training FLOPs than DFoT and DFoT-SSM to reach convergence, but remains within the same order of magnitude while delivering substantially more stable long-horizon predictions.

Inference wall clock time and memory usage is measured on 1 H200 GPU with two protocols, and is reported in Table 14: the first setting generates 1 frame with 1 frame of context, and the other generates 70 frames with 70 frames of context

*Table 13.* **Training compute on 3D Dynamic Block World.** FLOPs are measured for the forward pass and backward pass with batch size 1 and with 2 frames as reported in the first two columns. To compute the total training compute, we measure FLOPs at batch size 1 and with 140 frames, then scale by the actual batch size and number of training steps.

| Model | Forward Pass GFLOPs | Backward Pass GFLOPs | Batch Size | Steps | Total compute (EFLOPs) |
|---|---|---|---|---|---|
| FloWM (Ours) | 31.18 | 99.50 | 16 | 150k | 27.5 |
| DFoT | 4.94 | 9.91 | 32 | 300k | 15.7 |
| DFoT-SSM | 5.00 | 10.03 | 32 | 300k | 10.5 |

*Table 14.* **Inference scaling of rollout time and memory usage based on number of generated and context frames.** We measure the following on 1 H200 GPU with a batch size of 8. FloWM has the highest throughput and sublinear memory scaling with the number of generated and context frames.

| Model | Model size (M params) | Frames | Context | Rollout time (s) | Rollout throughput (generated frames/s) | Resident alloc (GiB) | Peak Delta (GiB) |
|---|---|---|---|---|---|---|---|
| FloWM | 12.8 | 2 | 1 | **0.015** | **522.6** | **0.190** | 0.314 |
| DFoT | 95.3 | 2 | 1 | 0.423 | 18.9 | 0.802 | **0.024** |
| DFoT-SSM | 97.8 | 2 | 1 | 0.801 | 10.0 | 0.810 | 0.393 |
| FloWM | 12.8 | 140 | 70 | **0.967** | **579.0** | **0.394** | **0.444** |
| DFoT | 95.3 | 140 | 70 | 3.353 | 167.0 | 1.024 | 1.698 |
| DFoT-SSM | 97.8 | 140 | 70 | 2.402 | 233.2 | 1.016 | 1.697 |

(matching the inference setting reported in the main text). For memory usage, in the table we report the resident allocated memory, which measures the GiB allocated before the generation begins, and the peak delta, which measures the additional memory allocated during the rollout above the resident allocated baseline. The rollout time for FloWM is significantly faster, and the memory usage scales sublinearly with the number of frames. These results intuitively follow from FloWM's instantiation as a recurrent network, contrasted with the combination of self-attention scaling from increasing the number of frames, and the repeated sampling procedure necessary for denoising with DFoT and DFoT-SSM. DFoT and DFoT-SSM can process context and generate frames in parallel with more frames in the window, which means the throughput improves with more generated frames, but it still lags behind FloWM's.

### I.2. FloWM Memory Map Hyperparameter Effects on Compute Efficiency

In Table 15, we additionally report the effect of the map and Field of View (FoV) size on memory allocation and rollout time. A larger grid and larger FoV naturally increases the rollout time and memory usage, though FloWM with these changes is still able to retain high throughput compared to the baselines.

### I.3. Limitations and Opportunities for Efficiency

Our current implementation is not heavily optimized, and there are several possible future avenues to reduce computational cost without changing the core modeling assumptions. First, decoupling the encoder from the hidden state update would allow processing of input frames in parallel before writing to the latent map, increasing input level parallelism. Second, designing the recurrent update to be linear and associative would, in combination with such a decoupling, make it amenable to parallel scan algorithms similar to those used in modern state-space models (Martin & Cundy, 2018; Smith et al., 2023). This would enable parallelization over sequence length while retaining a recurrent structure. Third, the latent map is currently updated via a dense gated operation over the entire estimated field of view, even though the true change in information per timestep is typically sparse. Introducing sparse or multiscale updates could substantially reduce per-step computation while preserving the benefits of a persistent world memory. Implementation could be inspired by other hierarchical mapping methods such as NICE-SLAM and SHINE-Mapping (Zhu et al., 2022; Zhong et al., 2023). Overall, these directions suggest that the additional compute required by FloWM during training is not fundamental to the architecture, but rather reflects design choices that can be systematically optimized in future work.

## J. RSSM Baseline Details

RSSMs, such as the architecture used in Dreamer V3, are popular choices for non-diffusion generative world models (Hafner et al., 2023). The RSSM learns an action-conditioned latent dynamics model for predicting future observations given action conditions. A convolutional encoder maps frames to observation embeddings, and the recurrent state-space model

*Table 15.* **Scaling of compute time and memory usage with FloWM map parameter choices.** With different latent map sizes and Field of View (FoV) choices for FloWM, we can see reasonable scaling of throughput and memory usage.

| FloWM setting | Map grid | FoV | FoV fraction | Decoder KV tokens | Map-update sequence tokens | Rollout time (s) | Rollout throughput (generated frames/s) | Resident alloc (GiB) | Peak Delta (GiB) |
|---|---|---|---|---|---|---|---|---|---|
| Default | $33 \times 33$ | $60°$ | 0.163 | 177 | 241 | 0.967 | 579.0 | 0.394 | 0.444 |
| Half map size | $17 \times 17$ | $60°$ | 0.183 | 53 | 117 | 0.606 | 923.9 | 0.394 | 0.317 |
| Larger FoV | $33 \times 33$ | $90°$ | 0.265 | 289 | 353 | 1.514 | 369.8 | 0.395 | 0.467 |

*Table 16.* **Additional model parameter size ablation on the RSSM baseline, on the 3D Dynamic Block World dataset.** Given 70 context frames, models are evaluated on generation of 70 (training objective) and 210 (length extrapolation) future prediction frames.

| Model | MSE ↓ | | PSNR ↑ | | SSIM ↑ | |
|---|---|---|---|---|---|---|
| | 70 | 210 | 70 | 210 | 70 | 210 |
| **FloWM (Ours)** | **0.000603** | **0.001539** | **32.19** | **28.13** | **0.9673** | **0.9525** |
| DreamerV3 RSSM (50M) | 0.01844 | 0.01881 | 17.34 | 17.26 | 0.8696 | 0.8673 |
| DreamerV3 RSSM (400M) | 0.01636 | 0.01647 | 17.86 | 17.83 | 0.8799 | 0.8782 |

maintains deterministic and discrete stochastic latent states. During training, posterior latents are inferred from the current observation and decoded back to pixel space with an MSE reconstruction loss. The objective also includes a dynamics KL term, which trains the prior dynamics to predict posterior latents, and a representation KL term, which regularizes posterior latents toward the prior. The dynamics model is repeatedly applied for long horizon rollouts. Our implementation is derived from the DreamerV3 codebase and includes 50M and 400M variants.

The results reported in the main text are using the 400M parameter variant; we report the rollout result comparison between the 400M and 50M variants in Table 16, and find that model scale does provide improvements, but they are not significant, reinforcing our hypothesis that structured memory is important for solving partially observed dynamic world modeling problems.

The models are trained on the Dynamic Block World split for 300k steps on a single H200 GPU. The models operate directly in pixel space at $128 \times 128$ resolution and consume 140-frame clips with 50 context frames. Actions are loaded as discrete Block World action IDs, aligned as $t - 1 \rightarrow t$ transitions, and converted to one-hot vectors before being passed to the RSSM. We use AdamW with learning rate 4e-5, betas (0.9, 0.99), weight decay 1e-3, constant learning rate schedule, and gradient clipping at 1.0. The RSSM objective combines pixel MSE reconstruction loss, dynamics KL, and representation KL with weights 1.0, 1.0, and 0.1 respectively, using 1.0 free nat for the KL losses, following the original paper. The training settings are detailed in Table 17, and the model settings are detailed in Table 18.

## K. Diffusion Baseline Details

### K.1. Video Diffusion Transformers

Diffusion Transformer based video generation models are the most prominent choice for video generation world models today (Peebles & Xie, 2023; Brooks et al., 2024). Training follows a similar formula with diffusion image generation pipelines, requiring attention over the temporal dimension to retain temporal consistency. For video data, diffusion models are typically trained within the latent space of a variational autoencoder (VAE) (Rombach et al., 2021; Gupta et al., 2024), where raw video frames are first compressed into a compact latent representation.

The ability for these video diffusion models to generate impressively realistic videos has led to an increased interest for their use as world models, and there is a growing focus in ensuring the spatiotemporal consistency of these models as world simulators. Due to the size complexity of the input token space, to generate long videos, researchers have turned to autoregressive sampling and sliding window attention; though ubiquitously used, we speculate that the drawbacks of this sliding window method for inference, where there is no hidden state passed between generation rounds after the window shifts, is a major reason that DiT baselines fail on the simple task presented in this work.

### K.2. Diffusion Forcing Transformer Baseline

Due to its claims of long term consistency and flexible inference abilities, for our baseline we chose a History-guided Diffusion Forcing training scheme, using latent diffusion with a CogVideoX-style transformer backbone, which we refer to as DFoT (Song et al., 2025; Chen et al., 2024; Yang et al., 2025; Rombach et al., 2021). Models for state of the art video

*Table 17.* RSSM training configuration for the Dynamic Block World dataset.

| Config | Block World |
|---|---|
| Effective batch size | 16 |
| Learning rate | 4e-5 |
| Weight decay | 1e-3 |
| Training steps | 300k |
| GPU usage | 1×H200 |
| Optimizer | Adam, betas=(0.9, 0.99) |
| Precision | FP32 |

*Table 18.* RSSM model configurations for the Dynamic Block World dataset.

| Config | 50M | 400M |
|---|---|---|
| Total parameters | 50 M | 400 M |
| Hidden dimension | 512 | 1536 |
| Deterministic recurrent units | 4096 | 12288 |
| CNN channels | 32 | 96 |
| Stochastic latent factors | 32 | 96 |
| Classes per latent factor | 32 | 32 |
| RSSM blocks | 8 | 8 |
| Activation | GELU | GELU |
| Norm | RMSNorm | RMSNorm |

world modeling today have similar training formulas and architectures for the backbone (Xiang et al., 2024; Ball et al., 2025; Decart et al., 2024; Agarwal et al., 2025). We first train a spatial downsampling VAE on frames of each dataset, then pass input video frames through the VAE to form a latent representation before it reaches the diffusion model.

Following the standard diffusion forcing training scheme, each frame during training is corrupted with independently sampled gaussian noise, and the training target is to predict some form of the ground truth from these noisy frames.

Specifically, during training, the full sequence is constructed as $\mathbf{x}_\tau = \{\mathbf{x}_t\}_{t=0}^{T-1}$, where each latent frame is assigned an independent noise level $k_t \in [0, 1]$. Each frame (more precisely, each collection of spatial tokens corresponding to a single frame) is noised according to the following equation:

$$\mathbf{x}_t^{k_t} = \alpha_{k_t} \mathbf{x}_t^0 + \sigma_{k_t} \epsilon_t, \quad \epsilon_t \sim \mathcal{N}(0, 1), \tag{31}$$

where $\alpha_{k_t}$ and $\sigma_{k_t}$ denote the signal and noise scaling factors, respectively, determined by the chosen variance schedule. The clean data frame is denoted as $\mathbf{x}_t^0$. The diffusion model $\epsilon_\theta$ takes in as input a sequence of noise levels, $k_\tau$, and the sequence of independently noised inputs $\mathbf{x}_\tau^{k_\tau}$. The model is trained to minimize the following denoising loss:

$$\mathbb{E}_{k_\tau, \mathbf{x}_\tau, \epsilon_\tau} \left[ \left\| \epsilon_\tau - \epsilon_\theta \left( \mathbf{x}_\tau^{k_\tau}, k_\tau \right) \right\|^2 \right]. \tag{32}$$

For more information on diffusion models in general, please see (Chan, 2024).

(Song et al., 2025) showed that using this training objective with per-frame independent noise allows for the history image frames to be prepended to the noisy frames as context in the same self attention window, with zero (or some minimal) noise level. This scheme where context frames are in the self-attention window, available to be used to condition generation of future frames, is called History Guidance.

For DFoT models, unlike FloWM recurrent models, during training we make no distinction between observation and prediction frames, and train on sequences of a particular length (70 for MNIST World, 140 for Block World) in the self-attention window, where each frame's tokens receive independent gaussian noise. During inference, we utilize History Guidance with a number of frames equal to the training length in the attention window, consisting of some context frames and some in-progress generated frames. For MNIST World the observation frames are given minimal noise, and the 20 prediction frames all begin at full noise; then the entire set of frames is passed through the model multiple times according to the scheduler to complete denoising the target frames, with the clean frames as the final output. Following the DFoT codebase, all the predicted frames have the same noise level that decreases after each denoising iteration. For Block World there are 70 context frames and 70 prediction frames.

### K.3. DFoT Training Details

We train a separate DFoT model for each MNIST World data subset to isolate the performance per subset, and another set of models for Block World. We embed actions using a simple MLP embedder, and concatenate it to the video tokens, following CogVideoX. Our 96M parameter DFoT's validation loss and validation metrics converge after 245k steps on 1 NVIDIA L40S 48GB GPU with a batch size of 128 on MNIST World, and after 300k steps on 2 NVIDIA L40S 48GB GPUs with a batch size of 32 on Block World. More training hyperparameters are reported in Table 19.

### K.4. DFoT-SSM Training Details

We train a separate DFoT-SSM model for each data subset to isolate the performance per subset. We embed actions using a simple MLP embedder, and concatenate it to the video tokens, following CogVideoX. Our 97.8M parameter DFoT-SSM's

*Table 19.* DFoT configurations for different datasets. Section and key are organized hierarchically in the first column.

| Config | MnistWorld | Block World |
|---|---|---|
| **Training** | | |
| Effective batch size | 128 | 32 |
| Learning rate | 2e-4 (linear warmup) | 2e-4 (linear warmup) |
| Warmup steps | 2,000 | 2,000 |
| Weight decay | 1e-3 | 1e-3 |
| Training steps | 245k | 300k |
| GPU usage | 1×L40S | 2×L40S |
| Optimizer | Adam, betas=(0.9, 0.99) | Adam, betas=(0.9, 0.99) |
| Training strategy | Distributed Data Parallel | Distributed Data Parallel |
| Precision | Bfloat16 | Bfloat16 |
| **Diffusion** | | |
| Objective | $v$-prediction | $v$-prediction |
| Sampling steps | 50 | 50 |
| Noise schedule | cosine | cosine |
| Loss weighting | sigmoid | sigmoid |
| **Model** | | |
| Total parameters | 95.3 M | 95.3 M |
| # attention heads | 12 | 12 |
| Head dimension | 64 | 64 |
| # layers | 10 | 10 |
| Time embed dimension | 256 | 256 |
| Condition embed dimension | 768 | 768 |
| **Inference** | | |
| History guidance | stabilized conditional (level = 0.02) | stabilized conditional (level = 0.02) |
| Context frames | 50 | 70 |
| Sampler | DDIM | DDIM |

validation loss and validation metrics converge after 200k steps on 2 NVIDIA L40S 48GB GPU with an effective batch size of 128 on MnistWorld, and 300k steps on 2 NVIDIA L40S 48GB GPU with an effective batch size of 32 on Block World. More training hyperparameters are reported in Table 20. DFoT-SSM is trained with a certain number of clean frames in the context (50 for MNIST World and 70 for Block World), and the remaining frames are noised in the same way as for DFoT.

For the Block World dataset, we allow all models to have 140 frames available during training, but the models use them differently. FloWM uses 50 context frames and calculates the loss by predicting the next 90. For the sliding window inference to work well for the DFoT-SSM, we chose 70 frames to match the training task and so that the sliding window could cleanly slide forward by 70 frames. To match the information given to each model based on this training setting for DFoT-SSM, we evaluate all the models with 70 frames of context, despite the fact that FloWM was trained with 50 frames of context only. We didn't find a significant difference in results between the two settings, but report with 70 frames of context for all models for fairness, as this is what matched the DFoT-SSM training setting. We also experimented with an autoregressive scheme where only one frame is predicted at a time instead of all the future in parallel, which produced similar results for both the DFoT and DFoT-SSM baselines.

### K.5. VAE Training Details

Following standard practice, we use a VAE to perform latent diffusion; doing diffusion on pixels instead could offer perceptually different results, but we do not believe it would alter the results of the model. We train our 8x spatial downsampling VAE on sample frames from a mix of all of the MNIST World data subsets, such that all combinations of overlapping MNIST digits are within the training distribution. Our 20M parameter VAE's validation loss converges at about 90k steps for MNIST World, using an effective batch size of 256 across 4 NVIDIA L40S 48GB GPUs with a learning rate of 4e-4. We utilize a Masked Autoencoder Vision Transformer based VAE (He et al., 2022). We directly apply the VAE code from Oasis (Decart et al., 2024), including an additional discriminator loss that helps with visual quality; please refer to their work for more details. The reconstruction MSE accuracy reaches 0.02 for MNIST World, so any DFoT MSE can be expected

to be 0.02 higher than if trained on pixels; we believe this should not affect convergence behavior of the DFoT models on the downstream task, since the diffusion model only ever sees the latent space. During diffusion training, for our MNIST World VAE with latent dimension 4, and spatial downsampling ratio 8, input videos of shape `[num_frames, channels, height, width]` are converted to shape `[num_frames, 4, height // 8, width // 8]`. For Block World, the MSE for the 80M parameter VAE is less than 0.003, and we train the model for 300k steps with a latent dimension 8 and spatial downsampling ratio of 16. More training hyperparameters are reported in Table 21 and Table 22.

*Table 20.* DFoT-SSM configurations. Classifier-free guidance (Ho & Salimans, 2022) for conditioning is not used during inference; though the models have been trained to allow for it, we find their instruction following ability not to be a limiting factor. Loss weighting uses sigmoid reweighting proposed by Kingma & Gao (2023) and adopted by Hoogeboom et al. (2025). History guidance follows the stabilized conditional method (level = 0.02) from Song et al. (2025); please refer to their codebase for details.

| Config | MnistWorld | Block World |
|---|---|---|
| **Training** | | |
| Effective batch size | 128 | 32 |
| Learning rate | 2e-4 (linear warmup) | 2e-4 (linear warmup) |
| Warmup steps | 2,000 | 2,000 |
| Weight decay | 1e-3 | 1e-3 |
| Training steps | 200k | 300k |
| GPU usage | 1×L40S | 2×L40S |
| Optimizer | Adam, betas=(0.9, 0.99) | Adam, betas=(0.9, 0.99) |
| Training strategy | Distributed Data Parallel | Distributed Data Parallel |
| Precision | Bfloat16 | Bfloat16 |
| Context frames | 50 | 70 |
| **Diffusion** | | |
| Objective | $v$-prediction | $v$-prediction |
| Sampling steps | 50 | 50 |
| Noise schedule | cosine | cosine |
| Loss weighting | sigmoid | sigmoid |
| **Model** | | |
| Total parameters | 97.8 M | 97.8 M |
| # attention heads | 12 | 12 |
| Head dimension | 64 | 64 |
| # layers | 10 | 10 |
| Time embed dimension | 256 | 256 |
| Condition embed dimension | 768 | 768 |
| **Inference** | | |
| History guidance | stabilized conditional (level = 0.02) | stabilized conditional (level = 0.02) |
| Context frames | 50 | 70 |
| Sampler | DDIM | DDIM |

*Table 21.* VAE configurations for MNIST World. The input size from the dataset is $32 \times 32$.

| Component | Option | Value |
|---|---|---|
| Training | Learning rate | 4e-4 |
| | Effective batch size | 256 |
| | Precision | Float16 mixed precision |
| | Strategy | Distributed Data Parallel |
| | Warmup steps | 10,000 |
| | Training epochs | 172 |
| | GPU usage | 4×L40S |
| | Optimizer (AE) | Adam, betas=(0.5, 0.9) |
| | Optimizer (Disc) | Adam, betas=(0.5, 0.9) |
| Model | Total parameters | 19.7 M |
| | Encoder dim | 384 |
| | Encoder depth | 4 |
| | Encoder heads | 12 |
| | Decoder dim | 384 |
| | Decoder depth | 7 |
| | Decoder heads | 12 |
| | Patch size | 8 |
| Latent | Latent dim | 4 |
| | Temporal downsample | 1 |

*Table 22.* VAE configurations for Block World. The input size from the dataset is $128 \times 128$.

| Component | Option | Value |
|---|---|---|
| Training | Learning rate | 4e-4 |
| | Effective batch size | 256 |
| | Precision | Float16 mixed precision |
| | Strategy | Distributed Data Parallel |
| | Warmup steps | 10,000 |
| | Training epochs | 40 |
| | GPU usage | 4×L40S |
| | Optimizer (AE) | Adam, betas=(0.5, 0.9) |
| | Optimizer (Disc) | Adam, betas=(0.5, 0.9) |
| Model | Total parameters | 80.7 M |
| | Encoder dim | 576 |
| | Encoder depth | 5 |
| | Encoder heads | 12 |
| | Decoder dim | 576 |
| | Decoder depth | 15 |
| | Decoder heads | 12 |
| | Patch size | 16 |
| Latent | Latent dim | 8 |
| | Temporal downsample | 1 |

