# OpenReview forum: "Flow Equivariant World Models: Structured Memory for Dynamic Environments"
_ICML.cc/2026/Conference — ICML 2026 regular_

### Official Review · Reviewer_yn62 · 2026-02-26

**Soundness:** 3
**Presentation:** 3
**Significance:** 3
**Originality:** 2
**Overall Recommendation:** 4
**Confidence:** 3

**Summary:**

Overall, this manuscript presents a pertinent issue: modern video world models often fail under partial observability + dynamics, because their “memory” is either a finite attention window or an unstructured cache that cannot evolve consistently when objects move out of view.
The manuscript's main contribution consists of Flow Equivariant World Modeling (FloWM): a framework that treats both self-motion and external object motion as Lie-group “flows” acting on a persistent latent world memory, and enforces (or induces) equivariance so the memory transforms in sync with world dynamics.

They instantiate FloWM with (i) a simple convolutional recurrent model with discrete “velocity channels” for 2D MNIST-World, and (ii) a ViT encoder/decoder with a spatial token-map memory and gated write-update for 3D Block-World, reporting strong long-horizon rollout stability and reduced hallucination compared to diffusion-forcing baselines.

**Compliance With Llm Reviewing Policy:**

Affirmed.

**Final Justification:**

The rebuttal clarified several of my main concerns and improved my confidence in the paper.
While some limitations remain, I believe the overall contribution and the substantial effort behind the work justify raising my score to 4.

**Key Questions For Authors:**

1. In the transformer FloWM, what is the exact implementation of the internal flow term for external object motion in token space, and how can a single map represent multiple objects with different velocities simultaneously without explicit velocity channels?

2. Please specify the group representations used in 3D Block-World: how exactly are translations and rotations applied to the latent map tokens (and any motion-related state), and what assumptions are required about the action space?

3. Why are comparisons restricted to DFoT / DFoT-SSM? Can you add strong non-diffusion world-model baselines (e.g., recurrent latent SSM/RSSM-style, action-conditioned transformer predictors, map-memory baselines) under matched compute?

4. What is the compute/memory scaling law of the latent map (token grid size, FoV fraction, sequence length)? Please report runtime/throughput and memory usage, not only rollout metrics.

**Limitations:**

- The method is evaluated under rigid-body geometric actions with known action parameterizations, and its current formulation leans on discrete/structured motion assumptions; extending to richer actions (semantic, deformable, continuous) is not demonstrated.

- The empirical validation is largely confined to synthetic simulators, limiting confidence in real-world egocentric dynamics and general robotics settings.

- The transformer version’s correctness relies on emergent encoder equivariance rather than guaranteed equivariant design, raising concerns about brittleness under distribution shift.

**Strengths And Weaknesses:**

__Strengths__

- The paper motivates partial observability failures with crisp schematic figures and a coherent narrative from “sliding-window” limitations to the need for structured memory..

- The progression from flow-equivariance definitions to an implementable recurrence and then to a transformer token-map design is generally well-organized, with explicit equations and an update rule that is easy to follow.

__Weaknesses__

- Novelty appears incremental relative to prior “map memory” and flow-equivariant RNN work. The paper heavily builds on flow-equivariant recurrence (cited as Keller 2025) and aligns closely with earlier spatial memory maps (e.g., Neural Map/EgoMap-style ideas). The new element is mainly a unifying group-theoretic framing plus an engineered memory update, but the “core mechanism” (structured shift/transform + write-to-map) is conceptually unsurprising.

- Experimental scope is too synthetic to substantiate claims of general dynamic-world modeling. MNIST-World and Miniworld Block-World are controlled toy simulators; even the “textured” variant remains far from real-world egocentric video with occlusions, non-rigid motion, lighting changes, and noisy actuation. This makes the empirical evidence insufficient for the strong generalization positioning.

- Baseline set is narrow and arguably misaligned with the evaluation objective. Comparisons focus on diffusion-forcing baselines (DFoT, DFoT-SSM). However, the proposed FloWM is trained with a deterministic MSE prediction loss (sequence regression), while diffusion models are generative and may require different evaluation protocols; moreover, key non-diffusion recurrent world models (RSSM/SSM-style predictive models, action-conditioned transformers without diffusion, map-based neural memories) are missing.

---

> ### Author Rebuttal · Authors · 2026-03-31
>
> # Response to Reviewer yn62
> We sincerely thank the reviewer for the thoughtful review, and for their comments identifying our setting as a “pertinent issue”. We will add all new results in the camera ready version. Below, we:
> - Add a compute-matched Dreamer V3-based RSSM baseline [1]
> - Include a new downstream MPC planning evaluation
> - Clarify novelty related to prior work, implementation questions, and add compute/memory details
>
> ## Non-diffusion baselines
> We include a new baseline: an autoregressive RSSM based on Dreamer V3 trained on the same Dynamic Blockworld dataset under matched compute. Please see the results table, and rollouts on the website: https://anonflowm.github.io/#rssm
>
> We find the RSSM model has similarly poor performance with the other baselines and is much worse than FloWM, suggesting our gains indeed come from addressing the world modeling limitation in Fig. 2. Rollouts show that the RSSM also fails to track out-of-view moving objects and instead defaults to average reconstructions, similar to the non-flow equivariant RNN baselines, rather than any evaluation artifact.
>
> ## Additional Evaluation: Downstream Planning
> We have also added a downstream training-free planning task, “Follow the Red Block,” where reward is the fraction of the image occupied by red pixels (a visual proxy for proximity to the red block).
> FloWM quickly approaches the red block even from distant initial states, while the baselines fail to do so. The same trend appears across 8 random episodes; see a distance over time figure, table, and pseudoalgorithm here: https://anonflowm.github.io/#planning
>
> ## Novelty relative to prior work
> Beyond Keller (2025) and Neural Map / EgoMap, our main contributions are:
> - Self-motion flow equivariance within a generalized flow equivariant recurrent relation (Eq. 4), extending structured memory to broader self / external group equivariance
> - A conceptual and empirical link between flow equivariance and memory in partially observed environments
> - Evaluation in dynamic world modeling
> - A ViT-based flow equivariant world model architecture that empirically learns an implicit mapping from 3D perspective observations to a top down 2D map
>
> ## Map-based memory baselines
> EgoMap-style baselines require explicit depth and camera geometry, which we intentionally avoid. Moreover, EgoMap is not a generative model: it uses a projected spatial memory for policy optimization in RL, making it not directly comparable in our setting. However, our “no velocity channels” ablation is conceptually close to a memory map baseline: it retains the self-motion shifted map memory but removes external-motion flow equivariance, and performs much worse than FloWM on both datasets.
>
> ## More realistic environments and learned equivariance
> We agree the current experiments are synthetic and should not be over-generalized; we will soften this and expand the limitations in the camera-ready. That said, the framework is more general than the datasets in this paper. It requires a latent space whose transformations are homomorphic to agent actions and external motions, plausible in settings like robotics where actions or object motion are known. We comment on settings with unknown structure further in the response to xoAc (“Action settings”).
>
> Our framework can describe how to build flow equivariant world models with exact equivariance as well -- the current transformer is just one scalable instantiation. An exactly equivariant 3D transformer is possible in principle, but significantly more expensive due to the need for a 3D latent space and depth unprojection.
>
> ## Runtime, compute, and memory
> We report total compute in Table 12 of the appendix, per-step compute/runtime in Table 13, and add the requested memory tables at this link: https://anonflowm.github.io/#memory
>
> ## Implementation of the Transformer FloWM and group representations
> We provide further details in Appendix F. For the transformer FloWM, the internal flow for a forward action shifts the token map spatially in the inverse direction of the agent's motion. For rotation, the token map is inversely rotated, and the velocity channels are permuted accordingly so that velocity vectors rotate with the agent heading. The transformer model does include multiple velocity channels; the notation in Sec. 3.2 suppressed the ν index for brevity, which may have caused confusion, and we will make this explicit in the camera ready. Each velocity channel represents all objects sharing a particular velocity, with spatial locations for different objects.
>
> Regarding the action space, our experiments use discrete translations and 90° rotations for simplicity, but the framework only requires valid group representations aligned with the input transformation groups. Continuous translations / rotations could be handled, for example, with Fourier or steerable representations of the roto-translation group (e.g. see 'Steerable CNN' of Cohen & Welling, 2016).
>
> [1] Hafner et al., arXiv 2023

---

> > ### Author Rebuttal · Reviewer_yn62 · 2026-04-03
> >
> > Thanks for the detailed rebuttal. It addressed most of my concerns. Given the substantial effort behind this work, I would like to raise my score to 4.

---

> > > ### Author Response · Authors · 2026-04-06
> > >
> > > We would like to thank the reviewer for taking the time to review our rebuttal response, for increasing their score, and acknowledging the possible contribution of this work to the community at large. We will update the paper with all new results and updates mentioned in our rebuttal, and we are happy to answer any further questions if they arise.

---

### Official Review · Reviewer_xoAc · 2026-03-05

**Soundness:** 2
**Presentation:** 3
**Significance:** 3
**Originality:** 4
**Overall Recommendation:** 5
**Confidence:** 3

**Summary:**

The paper proposes FloWM, a world modelling framework attempting to model dynamic enviroments by learning a hidden memory and enforcing equivariance of the model predictions with respect to structured transformations of the input representing both objects' and agent's motion. The model encodes partial observations of the agent into a hidden map that undergoes a set of structured transformations and is decoded back into the observation space to form the future prediction. The paper considers two instantiations of the framework: an RNN-based model evaluated on a 2D environment of MNIST World and a transformer-based model on a 3D environment of Dynamic Blocks. In both cases the model consistently outperforms prior work in terms of the accuracy of future predictions.

**Compliance With Llm Reviewing Policy:**

Affirmed.

**Final Justification:**

The rebuttal has addressed all my concerns. As mentioned in the strengths, I think the paper attempts at solving an important problem in the world modelling domain and demonstrates a novel perspective on temporal consistency and long context adherence.

**Key Questions For Authors:**

1) It appears that on the considered datasets, just a simple RNN-like baseline (FloWM with no SME and no VC) outperforms the prior work. Could the authors comment on this? Also, have the authors tried to increase the capacity of the RNN baseline? Intuitively, in the absence of the "hardcoded" object dynamics the network has to spend some of its compute to learn those. Including more baselines in the comparisons would definitely make the contribution stronger.
2) It would be interesting to see how well the memory scales with respect to the number of objects in the environment.
3) Could the authors clarify how they picture extending the method to more challenging settings (most importantly to unknown and time-dependent actions)? This would be important to adequately assess the potential impact of the work on the field.

**Limitations:**

yes

**Strengths And Weaknesses:**

### Strengths

1) The paper successfully models contiuous motion of other agents that leave the field of view of the ego-agent. Solving this problem is an important step towards temporally consistent world models that support large contexts.
2) The method is grounded in the framework of flow equivariant models, is sound and theoretically justified.
3) FloWM consistenty outperforms prior work across instantiations and datasets.

### Weaknesses

1) The main weakness of the proposed method is that it seems to strogly rely on the prior knowledge of possible object motions that are "hardcoded" in the transformations the hidden map undergoes. The authors mention in the discussion section that their future plans include extending the framework to more complex and potentially learned actions. However, the current framework does not seem to allow straight-forward scaling to more complicated environments and action spaces.
2) The evaluations are limited to overly simplified environments, with objects undergoing constant velocity motion.
3) The comparisons are limited to only two diffusion-based baselines that are outperformed by a simple RNN-based model.

---

> ### Author Rebuttal · Authors · 2026-03-31
>
> # Response to Reviewer xoAc
>
> We thank the reviewer for the thoughtful review, particularly for noting that our work is “sound, theoretically justified”, and that our problem is “an important step towards temporally consistent world models”.
>
> Below, we:
> - Include additional non-diffusion baseline results for a RSSM based on Dreamer V3 and show FloWM performs better [1]
> - Clarify why the simple RNN baseline can appear stronger than other methods when using pixel-based metrics such as PSNR
> - Clarify how FloWM extends to more complex settings, including non-constant velocity motion, time-dependent actions, and unknown actions
>
> ## Additional baseline results and evaluations
> We thank the reviewer for pointing out how we can make our results more complete. In response we have implemented an autoregressive state-space model based on Dreamer V3, and evaluated it on the same Dynamic Blockworld dataset. Please see the results table, and rollouts on the website: https://anonflowm.github.io/#rssm
>
> We find the RSSM model has similarly poor performance with the other baselines and is much worse than FloWM, suggesting our gains indeed come from addressing the world modeling limitation in Fig. 2. Rollouts show that the RSSM also fails to track out-of-view moving objects and instead defaults to average reconstructions, similar to the non-flow equivariant RNN baselines.
>
> We also include new planning evaluations in our responses to reviewers ijpk and yn62 and https://anonflowm.github.io/#planning, further showing the downstream utility of FloWM.
>
> ## Extension to more complex environments
> We appreciate the opportunity to clarify scalability. Visual complexity is largely handled by the encoder; Appendix D.2 suggests this is orthogonal to the flow equivariant memory. Therefore we expect stronger representation backbones to be able to extend the approach to richer environments.
>
> For external motion, Blockworld already includes non-constant velocity in the form of wall bounces, where objects reverse velocity multiple times during long rollouts. More generally, predictable changes can be handled via velocity mixing (Keller, 2025; Sec. E.5.2, Table 4), modeling effects like collisions, friction, or acceleration as input-dependent transfers between velocity channels. When motion is not deterministically predictable, FloWM would require a distributional extension to capture uncertainty, and as would world models. We intentionally focus on the deterministic case, but agree that distributional uncertainty modeling is an exciting direction.
> ## Action settings
>
> For agent motion, both datasets already include time-dependent actions (Sections 4.2, 4.3): the agent follows a biased random walk, so motion is non-constant and unpredictable. Self-motion equivariance uses the known action between timesteps to transform the latent state directly, allowing arbitrary action sequences without additional velocity channels.
>
> In this work, we assume that the actions are known. If the dataset has unknown actions, one could annotate it from frame pairs with an inverse dynamics model (e.g. VPT [2]). If the action is known, but the latent representation is unclear, one can impose a latent transformation structure that is homomorphic to the world action (as in Blockworld) and let the encoder learn the mapping into that space. If even that structure is unknown, a plausible extension is to learn action-conditioned latent operators in a JEPA-style framework. We view this as promising future work that is actively being explored [3].
>
> ## Memory scaling
> FloWM does not incur additional memory cost as the number of objects grows: objects with the same velocity are tracked jointly in the corresponding velocity channel and spatial location. The main bottleneck for scaling the number of objects is therefore the learned representation capacity, but we already see that the model is capable of tracking up to 10 objects in Dynamic Blockworld.
>
> ## Understanding the simple RNN-like baseline
> We agree the simple RNN-like ablation can appear surprisingly strong under the reported metrics, but we believe the interpretation should be that all the baselines are poor, but in different ways. In MNIST world, the simple RNN model often predicts all black, while the diffusion baselines hallucinate MNIST-like digits after losing track of the out-of-view objects. We introduce an additional figure to explain how the all-black prediction can lead to a better metric score compared to a plausible but wrong hallucination, at this link: https://anonflowm.github.io/#baseline-metric
>
> Predicting the walls correctly in Blockworld is analogous to predicting all black in MNIST world. However, rollouts (Figures 5, 6, website), and the new planning evaluations, demonstrate that only FloWM accurately can track object locations outside of the field of view; all other baselines fail in different ways.
>
> [1] Hafner et al., arXiv 2023
>
> [2] Baker et al., arXiv 2022
>
> [3] Dinh and Deny, arXiv 2026

---

> > ### Author Rebuttal · Reviewer_xoAc · 2026-04-02
> >
> > Thank you for the rebuttal. The authors have addressed all my concerns.

---

> > > ### Author Response · Authors · 2026-04-06
> > >
> > > We would like to thank the reviewer for thoughtfully reviewing our rebuttal response, and for agreeing to raise the review score. Please indicate if any further concerns arise, and we will be sure to update the camera-ready version with all new results and edits mentioned in the review.

---

### Official Review · Reviewer_NfG3 · 2026-03-12

**Soundness:** 3
**Presentation:** 3
**Significance:** 3
**Originality:** 3
**Overall Recommendation:** 4
**Confidence:** 4

**Summary:**

In this paper a structured memory is presented by imposing an equivariance condition between the agent’s actions and the latent representations of the model. In the considered scenario, an agent moves through a partially observed environment. The authors notice that both internal and external motion can be considered as flows, thus they propose a flow equivariance.
To achieve that, the authors require that both encoder and update modules are flow-equivariant (eqs. 5,6). Surprisingly, this leads to a much simpler update relationship (eq. 4). Adding flow due to the agent’s action leads to the update equation in lines 190,191 (left column). The authors use this theoretical insight for their model update in eq. 7. The advantage of the proposed approach is that the equivariance relation is learned instead of being explicitly modeled.

They show that their algorithm leads to improved world modeling such that it achieves much improved performance in long sequences. The algorithm is evaluated on two environments, one based on MNIST, the other on Miniworld. As can be seen in fig. 5, it achieves faithful modeling 150 steps into the future.

**Compliance With Llm Reviewing Policy:**

Affirmed.

**Final Justification:**

The authors have adequately answered all my questions in their rebuttal.

**Key Questions For Authors:**

- It might make the paper easier to read if the 2 sentences in lines 255-260 (left column) were presented in the introduction in a bit simplified form.

- line 156, right column, trivial lift: If E[f,h](v)=E[f,h](hat(v)), does that imply that v=hat(v)?

- line 164, left column: what is the “co-moving reference frame”?

**Limitations:**

Yes

**Strengths And Weaknesses:**

Replaces explicit equivariance relation modeling with learned relations; this is neat and can be handy for many problems. Considering the popularity of world models and robotic/agentic planning, the proposed idea could become influential.

---

> ### Author Rebuttal · Authors · 2026-03-31
>
> # Response to Reviewer NfG3
> We greatly appreciate the reviewer's comments on our paper, and specifically their emphasis on how our framework supports a 'learned equivariance relation'. We also believe that this is one of the greatest strengths of our approach, and we are grateful to the reviewer for highlighting it. Below, we provide clarifications based on the questions provided.
>
> ## Highlighting the potential for learned equivariance
> Following the reviewer's suggestion, in the camera ready, we will add the following sentence to the introduction to highlight that our framework supports learning equivariant encoders and decoders. We agree this makes the paper easier to read, and appreciate the reviewer's suggestion.
>
> "In settings where exact equivariance is difficult to specify directly in the original input space (such as in Figure 1), we demonstrate that we can enforce equivariant latent transformations with respect to self motion and this induces a learned equivariant map for world model encoders."
>
> ## Questions
> ### "If Ef,h=Ef,h, does that imply that v=hat(v)?"
>
> We thank the reviewer for highlighting the potential for confusion with this notation. In the text, for the definition of the generalized flow equivariant recurrence relation, we require the encoder to perform a 'trivial lift', informally defined as producing the same output for all velocity channels. Formally, we write this as: $E_\theta[f_t;h_t]{(\nu)}=E_\theta[f_t;h_t]{(\hat{\nu})}\forall\nu,{\hat{\nu}}\in\mathfrak{g}$. This means that the output of the encoder in channel $\nu$ (specifically, $E_{\theta}[f_t; h_t]{(\nu)}$) is equal to output of the same encoder in a different channel $\hat{\nu}$. With more than a single velocity channel, this means $\nu\neq\hat{\nu}$, so we are effectively tying the output channels together for the encoder. Note this is identical to how the original Flow Equivariant RNNs were constructed and is a core part of the flow equivariance proof. We will include this clarification in the main text.
>
> ### "what is the “co-moving reference frame?"
>
> We again thank the reviewer for this question. The terminology 'co-moving reference frame' is meant to provide intuition for the following equation in the text (Equation 2).
> $h_{t + \Delta t} = \sigma(\psi_{\Delta t}(\nu) \cdot h_t + f_t)$
> The basic idea is that for a recurrent neural network to process an input in a consistent manner regardless of motion, the hidden state of the network must simply move or transform in an identical manner to how the input is moving. In our notation, this movement or transformation of the hidden state is denoted by the flow $\psi_{\Delta t}(\nu)$ acting on the hidden state.
> This is often referred to as a co-moving reference frame (the hidden state itself is co-moving with the input, assuming an input flow of $\psi_{\Delta t}(\nu))$, and thus we adopt the term and associated intuition here. The original flow equivariance work (Keller, 2025) cited at the end of that sentence gives a more detailed description of what a co-moving reference frame means in a recurrent neural network.
>
> In the world modeling setting, having a latent state that is co-moving with the agent, means that it is effectively analogous to an egocentric map, following the agent as it moves around the world. However, since the latent state of the FloWM has multiple additional velocity channels, it can also be seen as simultaneously co-moving with external objects. In other words, the world model is processing the world and all objects in their corresponding associated moving frames of reference, effectively factoring out the motion transformations from the observation sequence and enabling the improved performance we demonstrate.
>
> ## Additional Results
> We hope these clarifications are satisfactory, and encourage the reviewer to see new results including a DreamerV3 RSSM baseline and planning experiments, which are explained in the other reviews, and with results available at these links
> https://anonflowm.github.io/#rssm
> https://anonflowm.github.io/#planning
>
> We are happy to additionally answer any further questions the reviewer may have.

---

> > ### Author Rebuttal · Reviewer_NfG3 · 2026-04-04
> >
> > I thank the authors for their additional explanations. I will raise my score.

---

> > > ### Author Response · Authors · 2026-04-06
> > >
> > > We thank the reviewer for taking the time to read through our rebuttal response and for indicating that they will increase the review score. **However it appears on our end that the review score has not changed, though the confidence has been modified.** We would appreciate it if the reviewer could take a moment to modify the score in the official review so that it is as clear as possible for the Area Chair to note. We are also more than happy to answer any further questions that the reviewer may have about the manuscript, and we will update the camera-ready version with any new results and the modifications we mentioned.

---

### Official Review · Reviewer_ijpk · 2026-03-13

**Soundness:** 4
**Presentation:** 3
**Significance:** 3
**Originality:** 3
**Overall Recommendation:** 4
**Confidence:** 3

**Summary:**

This paper introduces Flow Equivariant World Models (FloWM), a framework for partially observed dynamic world modeling that enforces equivariance to one-parameter Lie-group "flows" representing both self-motion and external object motion. The key insight is that in partially observed environments, requiring equivariance to world-state flows (not just observation flows) naturally induces a spatially structured persistent memory. The authors formalize a generalized flow equivariant recurrence relation (Equation 4) where the hidden state contains multiple "velocity channels" that flow according to their own vector fields. When the agent's actions are known, the model achieves self-motion equivariance by transforming the hidden state according to the latent group representation of actions, effectively building an allocentric "map" of the environment that updates consistently even for objects outside the current field of view. Experiments on 2D MNIST World and 3D Dynamic Block World show that FloWM outperforms diffusion-based baselines (DFoT, DFoT-SSM) in MSE, PSNR, and SSIM metrics, with particularly strong length generalization (e.g., training on 20 frames but evaluating on 150 frames). The authors also demonstrate reduced "equivariance error" and the ability to track object positions even when temporarily occluded.

**Compliance With Llm Reviewing Policy:**

Affirmed.

**Final Justification:**

We thank the authors for their rebuttal answers. Our concerns regarding real-world data evaluation or handling stochastic future uncertainty are considered out-of-scope by the authors. The authors propose to "could design a latent space with SO(N) flow equivariance parameterized by the action signals". We think this should be tested to make a significant contribution. More generally, we still think the contributions listed by the authors compared to other previous works that rely on FoV to build a memory of the past like Keller et al. 2025, are not significant. Hence, we did not raise our score.

**Key Questions For Authors:**

1. **Comparison to non-diffusion baselines**: How does FloWM compare to autoregressive world models (e.g., RSSMs from the Dreamer series) or latent prediction models (e.g., JEPA-WMs)? Given that actions are provided, is the stochasticity modeled by diffusion necessary, or would deterministic rollouts suffice?

2. **Extension to realistic environments**: The two environments (MNIST World and Block World) have explicit symmetry structure that enables the equivariant architecture design. How would you extend FloWM to real-world robotics scenarios, such as manipulation with moving camera viewpoints or egocentric navigation? Evaluation on simulators like Robocasa, Libero, or Minecraft could help validate the generality of the assumptions.

3. **Relationship to approximate equivariant methods**: The Related Work section mentions that "recent work proposes to approach the goal of equivariant world modeling in a more approximate manner." How do the authors position FloWM relative to these more flexible approaches? What are the tradeoffs between strict and approximate equivariance?

4. **Homomorphism claim (Line 174)**: The statement that the hidden state "becomes spatially structured in a manner homomorphic to the structure of the world" is intriguing. Could you clarify whether this is a formal mathematical claim? If so, what precisely is the homomorphism?

5. **Novelty relative to Keller et al. (2025)**: Could you explicitly enumerate the contributions of FloWM compared to the closest prior work? It appears the main novelty is Self-Motion Equivariance, but a clearer delineation would help readers assess the incremental contribution.

### Minor Issues

- Line 163: "is equivalent to global the motion" contains a grammatical error (likely a missing or extra word).
- The DFoT acronym is used in Figure 2 before being defined in Section 4.1. Please introduce the acronym earlier or expand it in the figure caption.

**Limitations:**

Key limitations that should be acknowledged include:

1. **Synthetic environments with known structure**: Both experimental settings have explicit symmetry groups that enable architecture design. The approach may not straightforwardly extend to environments where the relevant symmetries are unknown or approximate.

2. **No downstream task evaluation**: The paper evaluates only prediction quality metrics. Demonstrating utility for planning, control, or other downstream tasks would strengthen confidence in the practical value of the learned representations.

3. **Deterministic setting assumption**: The experiments appear to use deterministic dynamics. How FloWM would handle stochastic environments or partial observability beyond simple occlusion is not addressed.

**Strengths And Weaknesses:**

### Strengths

- **Clear exposition of equivariant world models**: The paper provides a well-written and accessible introduction to flow equivariance in the context of world modeling. The mathematical framework is presented in a pedagogical manner, making the connection between Lie-group theory and structured memory intuitive even for readers less familiar with geometric deep learning.

- **Strong long-horizon prediction performance**: The experimental results demonstrate substantial improvements over baselines on long-horizon rollouts. The length generalization results (training on 20 frames, evaluating on 150) are particularly notable and suggest that the equivariant structure provides meaningful inductive bias for temporal extrapolation.

- **Principled approach to persistent memory**: The theoretical contribution linking equivariance constraints to the emergence of spatially structured hidden states is elegant. The formalization in Equation 4 provides a clean abstraction that could inspire future work on structured world models.

- **Demonstrated occlusion handling**: The experiments show that FloWM can maintain accurate predictions of object positions even when objects are temporarily outside the field of view, which is a desirable property for partially observed environments.

### Weaknesses

- **Limited baseline comparisons**: The baselines (DFoT and Diffusion Forcing State Space Model) are exclusively diffusion-based video generation models. The paper would benefit from comparisons to non-diffusion alternatives, such as autoregressive models (e.g., JEPA-WMs [1]) or recurrent state-space models (e.g., Dreamer [2], PlaNet [3]). Since actions are provided and the environments appear deterministic, the role of diffusion in handling uncertainty is not clearly motivated. A discussion of when diffusion is necessary versus when simpler autoregressive models would suffice would strengthen the positioning.

- **Narrow experimental scope**: The evaluation focuses exclusively on video prediction metrics (MSE, PSNR, SSIM) compared to ground-truth futures. Given that world models are often used for planning and control, demonstrating downstream applications such as goal-conditioned planning or counterfactual reasoning would strengthen the practical relevance of the contribution.

- **Synthetic environments only**: Both experimental environments (MNIST World and Dynamic Block World) are relatively simple synthetic settings with known structure. It remains unclear how well the approach would transfer to more complex, realistic scenarios where the underlying symmetry group may not be known a priori.

- **Contribution clarity relative to prior work**: The relationship to Keller et al. (2025), cited as the closest prior work, could be clarified. The introduction would benefit from an explicit list of contributions distinguishing FloWM from existing flow-based equivariant latent world models.

Besides the limited baseline diversity, narrow experimental scope (no planning tasks), and restriction to synthetic environments, I do think this paper's underlying ideas have strong potential, hence my accepting score of 4/6.

[1] What Drives Success in Physical Planning with Joint-Embedding Predictive World Models? Terver et al. 2026.

[2] Mastering Diverse Domains through World Models. Hafner et al. 2023.

[3] Learning Latent Dynamics for Planning from Pixels. Hafner et al. 2018.

---

> ### Author Rebuttal · Authors · 2026-03-31
>
> # Response to Reviewer ijpk
> We sincerely thank the reviewer for the thoughtful review. Below, we:
> - Include additional non-diffusion baseline results: RSSM model based on Dreamer V3 [1]
> - Include a new downstream planning task, “follow the red block”, and demonstrate significantly improved performance of our model over others
> - Provide additional explanations addressing downstream evaluation, realistic environments, prior work, and equivariant methods
> We will include the added results and clarifications in the camera ready.
>
> ## Non-diffusion baselines
> We thank the reviewer for bringing up this area in which our results could be made more complete and convincing. In response we have implemented an autoregressive state-space model based on Dreamer V3, and evaluated it on the same Dynamic Blockworld dataset. Please see the results table and rollouts on the website: https://anonflowm.github.io/#rssm
>
> We find the RSSM model has similarly poor performance with the other baselines and is much worse than FloWM, suggesting our gains indeed come from addressing the world modeling limitation in Fig. 2. We also agree diffusion is not necessary here: deterministic rollouts suffice for this deterministic environment, as FloWM shows.
>
> ## Downstream planning task evaluation
>
> We have added a downstream planning task, “Follow the Red Block,” where reward is the fraction of the image occupied by the red block (a visual proxy for being close to the red block). For fairness during rebuttal, we use training-free MPC that exhaustively searches a 3-step action horizon under each world model.
>
> FloWM quickly approaches the red block even from distant initial states, while the baselines fail to do so. The same trend appears across 8 random episodes; see a distance over time figure, table, and pseudoalgorithm here: https://anonflowm.github.io/#planning
> Given the accuracy of FloWM’s predictions (as seen in rollouts on the website and in Figure 6), as well as the latent-probe results in Figures 7, 9, and 10, the planning result is intuitive since the task relies on accurately predicting moving block positions across time.
>
> ## More realistic environments
>
> While it is true that the experiments are synthetic, we believe they require a nontrivial solution for mapping perception to memory suggestive of generally applicability to more realistic environments. In Blockworld, structure is imposed only on the 2D latent map, which transforms homomorphically with the agent’s motion in the world plane. Then, the model has to learn the mapping from observations into a structured latent space without explicit supervision.
>
> This is an instantiation of the broader recipe enabled by our framework: impose flow equivariance in a latent space whose structure is known or hypothesized, and let the model learn the mapping from perception to memory that satisfies it. For example, if a robotic arm has N rotational degrees of freedom, one could design a latent space with SO(N) flow equivariance parameterized by the action signals. When no such prior structure is available, one would need joint structure learning and approximate equivariance; this is exciting, but beyond our scope. In our response to Reviewer xoAc (“Unknown Actions”), we outline one possible route to accomplish this.
>
> ## Homomorphism claim
>
> Yes, this is a formal mathematical claim. The latent map transforms homomorphically with respect to the ground truth egocentric world/state transformation. Enforcing self-motion equivariance means any self-induced group transformation of the input yields the corresponding group transformation of the latent map. We will state this more carefully in the camera ready.
>
> ## Relationship to approximate equivariant methods
>
> FloWM provides an exact construction for world model flow equivariance. When that construction is available, equivariance need not be learned, reducing training burden and improving generalization (Figs. 5b,c). When exact equivariance is not analytically available, the framework still gives a blueprint for how latent spaces should be structured so that learned equivariant encoders can inherit similar benefits (Fig. 6b). Compared with regularization-based approximate methods, the resulting learning efficiency and generalization are less fragile because the structure is built into the model rather than only encouraged by the loss.
>
> ## Novelty relative to Keller (2025)
> Relative to Flow Equivariant Recurrent Neural Networks, our main contributions are:
> - A substantially generalized flow equivariant recurrence relation (Eq. 4);
> - Self-motion flow equivariance, the key world modeling advance over Keller (2025);
> - A conceptual and empirical link between flow equivariance and memory in partially observed environments; and
> - A ViT-based flow equivariant world model architecture that empirically learns an implicit mapping from 3D perspective observations to a top down 2D map.
>
> [1] Hafner et al., arXiv 2023

---

> > ### Author Rebuttal · Reviewer_ijpk · 2026-04-03
> >
> > We thank the authors for their rebuttal.
> > Our concerns regarding real-world data evaluation or handling stochastic future uncertainty are considered out-of-scope by the authors. The authors propose to "could design a latent space with SO(N) flow equivariance parameterized by the action signals". We think this should be tested to make a significant contribution.
> > More generally, we still think the contributions listed by the authors compared to other previous works that rely on FoV to build a memory of the past like Keller et al. 2025, are not significant.
> > Hence, we will not raise our score.
> >
> > Also, we encourage the authors to add in their camera-ready a mention of other world-models approaches we suggested in our first review.

---

> > > ### Author Response · Authors · 2026-04-06
> > >
> > > We thank the reviewer for taking the time to read our rebuttal and manuscript again.
> > >
> > > We agree with the reviewer that demonstrating the flexibility of the flow equivariant framework in more realistic domains is an exciting direction for future work, but we do not believe that this extension is necessary to establish the present contribution. We intend for our work to serve as a concrete building block toward architectures that can accurately represent and remember the dynamic world for world modeling or other related tasks.
> > >
> > > We would like to clarify that the prior work of (Keller 2025) does not study partially observed settings, especially in 3d world modeling from egocentric visual observations, nor the problem of maintaining a persistent memory that transforms consistently under changes in the agent's FoV. We therefore emphasize that our work is the first to combine these elements in a single framework:
> > > - Generalized flow equivariance recurrence relation in partially observed dynamic environments
> > > - Self-motion flow equivariance, enabling the memory to update consistently with agent motion and to preserve information about objects outside the current view
> > > - A Recurrent ViT based architecture that learns to map 3d perspective observations into a structured latent map for long-horizon prediction.
> > >
> > > We further demonstrate that this predictive framework is useful for downstream planning tasks, whereas other models prone to hallucination cannot support the same use as reliably. We hope these clarifications are helpful to the reviewer, and we are happy to answer any further conceptual questions if they arise. All new experimental results and clarifications mentioned during our rebuttal
> > >
> > > We will include the experimental results and clarifications mentioned during our rebuttal in the camera ready version, along with the additional related world model approaches specifically mentioned in the original review.

---

### Decision · Program_Chairs · 2026-04-30

**Decision:**

Accept (regular)

**Comment:**

The paper introduces Flow Equivariant World Models (FloWM), to model partially seen dynamics by imposing equivariance between latent representations and motion dynamics. Both egomotion and object motions are represented as Lie-group flows on latent memory, enabling consistent updates even when objects get out of the FoV. Reviewers generally describe the work as technically sound and clearly written. They also appreciate the elegance of the formulation that links equivariance constraints with spatial memory. Experiments show superior long-term prediction and object tracking compared to the evaluated baselines. The rebuttal strengthened the end result by adding baselines and a planning experiment. This said, the reviewers also had more negative criticism. One was limited novelty compared to earlier work on flow-equivariant models vs. spatial memory. Moreover, experiments are limited to rather simplistic synthetic scenes. The rebuttal was judged to not completely lift such concerns. Overall, the paper presents advances in the use of structured memory in world models. All reviewers are positive (3 WA, 1 A). The AC therefore  believes the paper should be accepted.